# Natural Polyether Ionophores and Their Pharmacological Profile

**DOI:** 10.3390/md20050292

**Published:** 2022-04-26

**Authors:** Valery M. Dembitsky

**Affiliations:** Centre for Applied Research, Innovation and Entrepreneurship, Lethbridge College, 3000 College Drive South, Lethbridge, AB T1K 1L6, Canada; valery.dembitsky@lethbridgecollege.ca or valery@dembitsky.com; Tel.: +1-403-320-3202 (ext. 5463); Fax: +1-888-858-8517

**Keywords:** bacteria, cyanobacteria, dinoflagellates, invertebrates, fungal endophytes, QSAR, anticancer

## Abstract

This review is devoted to the study of the biological activity of polyether ionophores produced by bacteria, unicellular marine algae, red seaweeds, marine sponges, and coelenterates. Biological activities have been studied experimentally in various laboratories, as well as data obtained using QSAR (Quantitative Structure–Activity Relationships) algorithms. According to the data obtained, it was shown that polyether toxins exhibit strong antibacterial, antimicrobial, antifungal, antitumor, and other activities. Along with this, it was found that natural polyether ionophores exhibit such properties as antiparasitic, antiprotozoal, cytostatic, anti-mycoplasmal, and antieczema activities. In addition, polyethers have been found to be potential regulators of lipid metabolism or inhibitors of DNA synthesis. Further study of the mechanisms of action and the search for new polyether ionophores and their derivatives may provide more effective therapeutic natural polyether ionophores for the treatment of cancer and other diseases. For some polyether ionophores, 3D graphs are presented, which demonstrate the predicted and calculated activities. The data presented in this review will be of interest to pharmacologists, chemists, practical medicine, and the pharmaceutical industry.

## 1. Introduction

Natural polyether ionophore antibiotics are amazing chemical structures that are potent antibiotics and belong to the larger family of naturally occurring ionophores. It is known that the term “ionophore”, first used in 1967, refers to the ability of a molecule to bind a metal ion and facilitate its transport through cell membranes. This chemical-physiological property has made polyether ionophores a useful tool in studying the mechanisms of cation transport and served as a rationale for their biological activity [1,2,3,4]. Polyether ionophores are a class of organic substances that are squalene derivatives with a regular occurrence of multiple C-O-C motifs, including the distinct families of the ladder polyethers and exhibiting a wide range of high biological activities [5,6,7]. However, according to the Nakanishi hypothesis, many marine polyether metabolites can be biosynthesized from polyunsaturated fatty acids [8,9]. This class of polyether compounds is produced by unicellular marine algae, red seaweeds, marine sponges, and coelenterates, or is accumulated by some marine organisms such as mollusks [10,11,12,13].

Marine polyether ionophores are also known as phycotoxins and are produced to a large extent by phytoplankton and, in particular, dinoflagellates and diatoms. These microalgae and cyanobacteria in aquatic ecosystems, due to the massive increase in their biomass, create a natural phenomenon known as harmful algal blooms [14,15,16]. Algal blooms in seas and lakes have a negative impact on aquatic ecosystems, given that many of the polyether metabolites are toxic, which destroy coastal flora and fauna and are fatal to humans and animals [17,18,19].

The review focuses on natural polyether compounds isolated from bacteria, cyanobacteria, dinoflagellates, fungi, and some marine invertebrates. Linear and cyclic polyether toxins are a class of natural compounds that exhibit a wide range of biological activities such as strong antitumor, antifungal, and antibacterial activities. A comparative characteristic of the pharmacological profile of the individual chemical structure of each polyether ionophore using the QSAR method is presented [20,21,22].

## 2. Polyether Metabolites Produced by Dinoflagellates

Okadaic acid is one of the first marine polyether metabolites that was discovered over 50 years ago. In the early 1970s, a series of food poisoning outbreaks in Japan are known to have led to the discovery of a new type of shellfish poisoning in Tokyo, Yokohama, and mussels and scallops harvested in Miyagi Prefecture [23,24,25,26,27,28,29,30]. Numerous studies of these mollusks have shown that these bivalves accumulate okadaic acid (**1**) and its derivatives (**2**–**5**) [23]. It turned out that okadaic acid is a polyether compound derived from the fatty acid C38 [24].

Toxins of the okadaic acid group (Figure 1) are produced by dinoflagellates of the genera *Prorocentrum* [30,31,32,33,34] and *Dinophysis* [35,36,37], while esters are produced by microalgae (*Dinophysis* and *Prorocentrum* spp., see Figure 2) [38,39,40,41,42] or by mollusks by esterification of these toxins. All of these toxins can lead to diarrheal shellfish poisoning [26].

According to published experimental data [30,31,32,33,34,35,36,37,38,39,40,41,42] (see Table 1), one of the main properties of okadaic acid is that it is a protein phosphatase inhibitor and an inhibitor of platelet aggregation. In addition, this acid demonstrates anticancer, antifungal, antiparasitic, and antimitotic activities. These activities were confirmed by PASS (*Prediction of Activity Spectra for Substances*) with a confidence level of 84 to 99%. Of greatest interest is the fact that okadaic acid can act as an inhibitor of DNA synthesis with more than 99% confidence. This valuable property of okadaic acid is reflected in Figure 3. Similar results were obtained for derivatives of okadaic acid with a reliability of 98.3 to 98.8% (Table 2).

It is known that okadaic acid exhibits various biological activities and these are shown in Table 1. All experimentally found activities were confirmed by PASS and the degree of significance of each activity is indicated. In this regard, several biological activities showed a reliability of more than 96%. We isolated the activity called DNA synthesis inhibitor with more than 99% confidence (3D graph seen in Figure 3). This function of okadaic acid was discovered over 30 years ago [30,31,32,33,34,35,36,37,38,39,40,41,42].

Murakami and colleagues, working with dinoflagellate blooms in a rocky basin on Jogashima Island in Japan, isolated and identified an antifungal toxin named goniodomin A (**6**, for structure see Figure 4) [43]. They identified the dinoflagellate as *Goniodoma pseudogoniaulax*, which is currently considered synonymous with *Alexandrium pseudogonyaulax*, but the organism was later revealed to be *A. hiranoi* [44,45], a morphologically similar species. Two additional *Alexandrium* species, *A. monilatum* and *A. pseudogonyaulax,* have since been found to produce goniodomin A [46,47,48,49]. 

In addition, goniodomin A exhibits antiangiogenic activities via inhibition of actin reorganization in endothelial cells [50]. Goniodomin A as an inhibitor of angiogenesis with 97.9% confidence and its 3D graph is shown in Figure 5. In addition, goniodomin A shows antifungal (92.2%) and antibacterial (90%) activities.

Azaspiracids (**7**–**9**, AZAs) are a group of lipophilic polyether toxins produced by dinoflagellates belonging to the genera *Azadinium* and *Amphidoma* and are found in the shellfish and tunicates that accumulate them [51,52,53,54]. The AZAs induced cytotoxic and neurotoxic effects; however, the mechanism of action is still unknown [55].

Marine polyether metabolites named pectenotoxins (PTXs, **10**–**16**, for structures see Figure 6, predicted activity shown in Table 3, and 3D graph is shown in Figure 7) are a group of toxins associated with diarrhetic shellfish (particularly scallop *Patinopecten yessoensis*) poisoning (DSP) and are isolated from DSP toxin-producing dinoflagellates. As demonstrated by numerous studies that DSP toxins are produced by several of the *Dinophysis* species, including *D. acuta, D. fortii, D. acuminata, D. norvegica, D. mitra* and *D. caudata* [56,57,58,59], they are also produced by benthic species such as *Prorocentrum lima* [60,61]. Thus, the PTXs cause severe acute diseases in humans, as these toxins are highly cytotoxic, promote tumor development, and cause hepatocyte necrosis. In addition, nothing is known about the chronic toxicology of PTXs or the potential long-term public health effects [62,63].

The unique structure of a toxin named prorocentin (**17**) possessed all-*trans* trienes, an epoxide, as well as the 6/6/6-*trans*-fused/spiro-linked polyether ring moieties, is produced by the marine benthic dinoflagellate *Prorocentrum lima*, which is widespread in ocean waters anywhere in the world [64]. Prorocentin exhibited inhibitory activity against human colon adenocarcinoma DLD-1 and human malignant melanoma RPMI7951 with IC_50_ values of 16.7 and 83.6 µg/mL, respectively [64]. 

Gambierol (**18**) is a marine polycyclic ether toxin that is produced by the dinoflagellate *Gambierdiscus toxicus* and is a potent blocker of voltage-gated potassium channels [65,66,67]. Another analogue of gambierol named gambierone (**19**, seen in Figure 8) was isolated from the cultured dinoflagellate *Gambierdiscus belizeanus*, as well as gambierone and 4-methylgambierone (**20**), which were found in another species, the benthic dinoflagellate *Gambierdiscus australes* [68,69]. Potential antifungal polyether compounds called gambieric acids A (**21**), B (**22**), C (**23**), and D (**24**, for structure see Figure 8) were isolated from extracts from the marine dinoflagellate *Gambierdiscus toxicus* [70].

A group of lipid soluble polyether compounds called ciguatoxins (CTXs, **25**–**27,** structures seen in Figure 9) are potent ichthyotoxins produced by a toxic benthic dinoflagellate, *Gambierdiscus toxicus* [71,72,73]. The main toxin, named ciguatoxin, was first isolated from the liver of a moray eel caught off the coast of the Hawaiian Islands. This epiphytic *G. toxicus* dinoflagellate (see Figure 10) is the main source of toxins that accumulate in various fish species [74,75]. Ciguatera fish poisoning is a form of food poisoning caused by the consumption of varieties of toxic ciguatera fish species from tropical and subtropical waters [76,77,78].

A single-celled phytoplanktonic organism, the marine dinoflagellate *G. toxicus,* is found in tropical waters around the world. Maitotoxin **(27** and **28**, for structure see Figure 9, predicted activity shown in Table 4, and 3D graph seen in Figure 11 and Figure 12) and related toxins accumulate in the food chain when predatory fish consume contaminated herbivorous reef fish. Maitotoxin accumulates mainly in the liver and internal organs of fish, but not in their flesh. Higher concentrations of toxins can be found in large carnivores such as barracuda, sea bass, amberjack, perch, and others [79,80].

A microscopic single-cell photosynthetic organism of the genus Karenia, *K. brevis* (see Figure 10b), is a marine dinoflagellate that is commonly found in the waters of the Gulf of Mexico and is responsible for the so-called *Florida Red Tides* (see Figure 13 and Figure 14) that affect the Florida and Texas Gulf coasts of the United States and the nearby coasts of Mexico. *Karenia brevis* produces a set of potent neurotoxins, collectively called brevetoxins (PbTxs, **29**–**34**, for structure see Figure 15, predicted activity shown in Table 4), that cause gastrointestinal and neurological problems in other organisms and are responsible for the massive death of marine organisms and seabirds [81,82,83].

A marine toxin and complex polycyclic ether called brevisulcenal-F (**35**, for structure see Figure 16) was found in an extract of the dinoflagellate *Karenia brevisulcata*, which was dominant in the red tide of *K. brevisulcata* in Wellington Harbour, New Zealand. An extract of *K. brevisulcata* showed potent mouse lethality and cytotoxicity, and laboratory cultures of *K. brevisulcata* produced a range of novel lipid-soluble toxins [84]. Other polycyclic ether toxins, namely brevisulcenals A1 (**36**) and brevisulcatic acids (BSXs, **37**–**39**, structures seen in Figure 17, and see 3D graph in Figure 18) produced by the red tide dinoflagellate *K. brevisulcata*, were the cause of a toxic incident that occurred in New Zealand [85].

Two cytotoxic polyethers named gymnocin A (**40**) and B (**41**) were isolated from the notorious red tide dinoflagellate, *Gymnodinium mikimotoi* [86,87,88]. 

Ciguatoxins (**42**–**45**, for structures see Figure 19, and activity see in Table 5) are polyether toxins derived from marine dinoflagellates. Ciguatera fish poisoning (CFP) is currently the most common marine biotoxin food poisoning worldwide, associated with human consumption of circumtropical fish and marine invertebrates that are contaminated with ciguatoxins. Ciguatoxins are a potent sodium channel activator and contain neurotoxins that pose a health hazard at very low concentrations [89,90].

The use of fish from the tropics, which are contaminated with ciguatoxins, leads to various diseases characterized by neurological, cardiovascular, and gastrointestinal disorders, and the toxins themselves demonstrate the ability to cause persistent activation of voltage-gated sodium channels, which increases the excitability of neurons and the release of neurotransmitters. There is currently no medically approved treatment for ciguatera [89,90,91].

The pinnatoxins A (**46**, for structure see Figure 20, predicted activity shown in Table 6), C (**47**), and D (**48**) were first found in a bivalve, the pen shell *Pinna muricata,* in Japan [92,93,94]. Other pinnatoxins (**49**–**53**) were detected in Pacific oysters in New Zealand, following the determination of pinnatoxins E, F, and G in shellfish from South Australia [95]. These toxins have been shown to be synthesized by related dinoflagellates of the genera *Ensiculifera, Pentapharsodinium*, and *Bysmatrum* [96,97]. 

An imine neurotoxin called pinnatoxin G acts through the antagonism of nicotinic acetylcholine receptors with preferential binding to the α7 subunit, often activated in cancer. Since the increased activity of the α7 nicotinic acetylcholine receptor promotes increased growth and resistance to apoptosis, the effect of pinnatoxin G on cancer cell viability was tested. In a panel of six cancer cell lines, all cell types lost their viability, but HT29 colon cancer cells and LN18 and U373 glioma lines were more sensitive than MDA-MB-231 breast cancer cells, PC3 prostate cancer cells, and U87 glioma cells, respectively with levels of expression of nicotinic acetylcholine receptors α7, α4, and α9 [98]. 

Yessotoxin (**54**, structure seen in Figure 21) is a disulfated polyether toxin produced by dinoflagellates and accumulated in filter feeding shellfish. This toxin was first isolated in 1986 in Mutsu Bay, Japan from digestive glands of scallops *Patinopecten yessoensis* after a food intoxication episode [99]. Later, *Protoceratium reticulatum, Lingulodinium polyedrum, Gonyaulax spinifera*, and *G. taylorii*, were identified as the dinoflagellates that produce yessotoxin and its analogues (**55**–**61**, structures seen in Figure 22, and 3D graph seen in Figure 23) [100,101]. In addition to Japan, this toxin has been identified in shellfish harvested in Europe, including Spain, Italy, Norway, the Adriatic Sea, and the Sea of Japan, Russia; Chile; and New Zealand [102,103]. Yessotoxin and its analogues can induce programmed cell death at nanomolar concentrations in different model systems [104].

The family of polyene-polyhydroxy compounds named amphidinols (**62**–**67**, see Figure 24) were isolated as a potent hemolytic and antifungal agent from a cultured strain of dinoflagellates *Amphidinium klebsii*, *A. carterae*, and *Amphidinium* sp. [105,106,107,108].

The karlotoxins are a class of amphidinol-like compounds (**68**–**72**, structures seen in Figure 25, predicted activity shown in Table 7, and 3D graph seen in Figure 26) produced by mixotrophic strains of the dinoflagellate *Karlodinium veneficum* (originally *Gymnodinium veneficum*), and *Amphidinium* sp. The karlotoxins have been reported to display a variety of interesting effects on biological systems, including cellular lysis, damage of fish gills, and immobilization of prey organisms. The cytolytic activity of the karlotoxins is modulated by membrane sterol composition, which has been proposed as a mechanism for *K. veneficum* avoiding autotoxicity. This dinoflagellate has been implicated in several fish kill events apparently caused by the damaging effects of the karlotoxins [109,110,111,112].

## 3. Polyether Ionophores Derived from Marine Algae and Invertebrates 

Marine invertebrates and seaweeds are the source of many bioactive compounds, including unusual or rare lipids, fatty acids, and, of course, phycotoxins [113,114,115,116,117,118,119,120,121,122]. Thus, the golden alga, *Prymnesium parvum* is a haptophyte species that produces phycotoxins called prymnesin 1 (**73**, structure seen in Figure 27, predicted activity shown in Table 8) and prymnesin 2 (**74**), usually during red tide algal blooms. These toxins mostly kill fish and appear to have little effect on cattle or humans. Although harmful effects to humans are unknown, it is not recommended to consume dead or dying fish that have been exposed to *P. parvum* blooms [123,124,125,126]. 

Palytoxin (**75**, seen in Figure 28) was originally isolated in Hawaii from the tropical soft coral *Palythoa* sp., the zoanthids, and is produced by dinoflagellates from the genus Ostreopsis (*Ostreopsis siamensis*, *O. mascarenensis, O. lenticularis*, and *O. ovata*) [127,128,129,130]. Although palytoxin was first discovered in the extract of *Palythoa* spp., this toxin was also found in organisms living in close association with colonial zoanthids (*Palythoa caribaeorum*, *P. tuberculosa,* and *Palythoa* sp.) [129,131]. In addition, the toxin and its analogs (**76**, 3D graph seen in Figure 29) have been found in many marine organisms: *Artemia salina* (brine shrimp), *Equinometra lucunter* (rock boring urchin), *Haliotis virginea* (sea snail), *Evechinus chloroticus* (New Zealand sea urchin), *Pecten novaezealandiae* (N.Z. scallop), *Crassostrea gigas* (Pacific oyster), *Perna canaliculus* (green-lipped mussel), and *Mytilus galloprovincialis* (Mediterranean mussel) [132], as well as primary producers such as the red algae *Chondria crispus* and *Ch. armata* [133,134] and benthic dinoflagellates *Ostreopsis* spp. [135,136,137,138]. In addition, bacteria associated with antecedent organisms have also been studied as a possible source of this toxin production. This is supported by the fact that the hemolytic activity of the toxin was found in extracts of bacteria such as *Pseudomonas*, *Brevibacterium*, *Acinetobacter, Bacillus cereus*, *Vibrio* sp., and *Aeromonas* sp. [139,140,141]. Thus, the presence of palytoxin and analogs in various marine organisms may indicate the bacterial origin of this toxin [142].

## 4. Polyether Ionophores Produced by Actinomycetes

Genus Streptomycetes (family Streptomycetaceae, Gram-positive bacteria) produce a wide variety of commercially important polyketide compounds, including the well-known unusual fatty acids, macrolide, polyene, and polyether antibiotics, which exhibit antibacterial, antifungal, anthelmintic, antitumour, and immunosuppressive activities [143,144,145,146,147,148,149,150].

Terrosamycins A (**77**, structure seen in Figure 30, predicted activity shown in Table 9, and 3D graph seen in Figure 31) and B (**78**), two polycyclic polyether natural products, were purified from the fermentation broth of *Streptomyces* sp. RKND004, isolated from Prince Edward Island sediment. Like other polyether ionophores, both compounds exhibited excellent antibiotic activity against Gram-positive pathogens. Interestingly, the terrosamycins also exhibited activity against two breast cancer cell lines [151]. Two polyether-type metabolites (**79** and **80**) were isolated from the marine-derived *Streptomyces cacaoi* and showed antimicrobial activity, while (**80**) also showed anticancer activity against Hela, PC-3, and A549 [152].

A polyether antibiotic named ionomycin (**81**) with a high affinity for calcium ions was obtained in pure form from fermentation broths of *Streptomyces conglobatus*. Ionomycin is a narrow spectrum antibiotic that is active against Gram-positive bacteria such as *Staphylococcus aureus* FAD 209P, *Streptococcus pyogenes* C 203, *Bacillus subtilis* ATCC 6633, *Micrococcus luteus* ATCC 9341, *Diplococcus pneumoniae* ATCC 6303, *Corynebacterium diphtheriac* ATCC 19401, and *Clostridium tetazomorplum* SC 3103 [153,154,155,156,157].

Other antibiotics called salinomycin (**82**, 3D graph seen in Figure 32), SY-2 (**84**), and SY-9 (**85**) were produced by a strain of *Streptomyces albus* (ATCC 21838) [158,159,160]. Salinomycin is active against Gram-positive bacteria including mycobacteria and some filamentous fungi: *Bacillus subtilis* PCI 219, *B. cereus* IFO 3466, *B. circulans* 1170 3329, *B. megaterium* IFO 3003, *Staphylococcus aureus* FDA 209P, *S. aureus, S. epidermidis* IFO 3762, *Sarcina lutea* NIHJ, *Micrococcus flavus* IFO 3242, *M. luteus* IFO 2763, *Mycobacterium smegmatis* ATCC 607, *M. phlei* IPCR, and *M. avium* IFO 3153. Narasin (**83**) is also an antibiotic polyether and is produced by a strain of *Streptomyces aureofaciens*. It differs from salinomycin in an extra methyl group in position 4. Narasin is active in vitro against Gram-positive bacteria, anaerobic bacteria, and fungi and is effective in protecting chickens from coccidial infections: *Ceratocystis ulmi, Mycoplasma gallisepticum, M. hyorhinis, M. synoviae, M. hyopneumoniae, M. hyosynoviae*, *Trichophyton mentagrophytes, Actinomyces bovis, Clostridium innocuum, C. perfringens, Eubacterium aerofuciens, Peptococcus anaerobius, Propionibacterium acenes, Fusobacterium symbiosunt*, and *Bacteroides fragilis* [161,162]. In addition, narasin is active against several viruses including vaccinia virus, herpes virus, type III poliovirus, transmissible gastroenteritis, Newcastle disease virus, and infectious bovine rhinotracheitis virus [162].

Two isomeric homologs of lasalocid A (**86**, structure seen in Figure 33, predicted activity shown in Table 10) and B (**87**) have been isolated from cultures of *Streptomyces lasaliensis*. The homolog B differs from lasalocid A in that at position 4 of the benzene ring, homolog B contains an ethyl group, and homolog A contains a methyl group [163,164]. Both homologues are known to exhibit anticoccidial activity [165].

A polyether antibiotic named tetronomycin (**88**, 3D graph seen in Figure 34) and an acetylated derivative (**89**) were found in the extract of a strain of *Streptomyces* sp. *nov*. Tetronomycin sodium salt shows a broad antibiotic activity against all Gram-positive bacteria tested and is also active against several *Mycoplasma laidlawii* and *Neisseria pharynges* species. Activity against other Gram-negative bacteria is lacking as well as an inhibition of yeasts and filamentous fungi: *Staphylococcus aureus, Streptococcus faecalis, Micrococcus lysodeikticus, Bacillus subtilis, Micrococcus luteus*, and *Clostridium pasteurianum* [166]. 

Noboritomycins A (**90**) and B (**91**), two ionophoric polyethers, were isolated from a strain of *Streptomyces noboritoensis*. An unusual spiroketal system as well as a salicylic acid chromophore represent further remarkable elements. Noboritomycin A shows, in this respect, structural relationships to both salinomycin and lasalocid. Comparison of physico-chemical data, in particular the interpretation of the 1H- and 13C-NMR spectra, revealed that noboritomycins A and B are structurally closely related, with noboritomycin B carrying an ethyl substituent on the aromatic ring in the place of a methyl group present in noboritomycin A. Both metabolites exhibit activity against Gram-positive bacteria *Staphylococcus aureus, Micrococcus lysodeikticus, Micrococcus sp., Bacillus subtilis, Streptococcus faecalis, Sarcina lutea, Neisseria pharynges, Clostridican pasteurianum*, and *Mycoplasma laidlawii*, and against *Eimeria tenella* (chicken coccidiosis) [167].

An antibiotic named lysocellin (**92**, or K-5610) was isolated from *Streptomyces cacaoi* var. *asoensis* K-9 Met-. It had antimicrobial activity against Gram-positive bacteria, antibiotic-resistant *Staphylococcus aureus* and some fungi, but not against Gram-negative bacteria [168]. 

The molecular structure and the cation binding of nigericin (**93**), an antibiotic affecting ion transport and ATPase activity in mitochondria, has been detected in *Streptomyces* sp. The molecule is found to be like monensin, another antibiotic of similar properties [169]. Two years later, an antibiotic related to nigericin was named grisorixin (**94**), isolated from cultures of a strain of *Streptomyces griseus* [170]. It shows microbial activity against Gram-positive bacteria and fungi: *Bacillus subtilis* CIP 5262, *Staphylococcus aureus* CIP 53156, *Streptococcus pyogenes* CIP 561, *Mycobacterium chelonii* CLA 1952, *Streptomyces antibioticus* CLA 3430, *Saccharomyces cerevisiae* CLA 15, *Madurella mycetoni* CLA 1313, *Penicillium roqueforti* CLA 1617, *Aspergillus ochraceus* CLA 1714, *Endothia parasitica* CLA 516, *Cercospora beticola* CLA 32, *Rhizoctonia solani* CLA 1718, *Phoma betae* CLA 162, *Sclerotinia sclerotiorum* CLA 183, *Monilia laxa* CLA 1312, *Phomopsis mali* CLA 1613, *Botrytis cinerea* CLA 23, *Verticillium albo-atrum* CLA 211, *Epichloe typhina* CLA 519, *Helminthosporium festucae* CLA 76, *Dactylium dendroides* CLA 44, *Trichothecium roseum* CLA 1810, *Colletotrichum lindemuthianum* CLA 311, and *Ascochyta pisi* CLA 116 [171].

The polyether antibiotic, X-206 (**95**), was isolated from *Streptomyces* sp. strain K99-0413 and has shown potent antimalarial properties in vitro against drug-resistant *Plasmodium falciparum* [172,173,174]. The pandemic spread of new human pathogenic viruses, such as the current SARS-CoV-2, is a major health and social concern. Antibiotic X-206 has shown potent ability to inhibit SARS-CoV-2 replication and cytopathogenicity in cells. Thus, the antibiotic X-206 can be considered one of the reliable agents for the treatment and prevention of SARS-CoV-2 coronavirus [175]. An antibiotic polyether with anthelmintic properties, A204A (**96**), has been found in the culture of *Streptomyces albus* [176].

Septamycin (**97**) is a metal-complexing polyether antibiotic produced by a strain of *Streptomyces hygroscopicus* NRRL 5678. Septamycin possesses antiviral activity against Newcastle Disease and herpes simplex viruses. In addition, this antibiotic in vitro shows antimicrobial activity against: *Staphylococcus aureus, Streptococcus pyogenes, Bacillus subtilis, B. stearothermophilus, Escherichia coli*, and *Clostridium pasteurianum* [177]. The ionophores septamycin (**97**), salinomycin (**82**), and CP-82,009 were purified from fermentation broth of the *Actinomadura* sp. culture N742-34 [178,179]. The other two are polyether antibiotic K-41A (**98**, structure seen in Figure 35, predicted activity shown in Table 11) and its analogue K-41Am (**99**), with antibacterial activity against Gram-positive bacteria and coccidia and anti-HIV activity exhibited, produced by a marine-derived *Streptomyces* sp. SCSIO 01680 [180].

An antibiotic, macrocyclic lactone carbonic acid named sorangicin A (**100**) was found in the culture supernatant of the myxobacterium *Sorangium (Polyangium) cellulosum* strain eel2. The antibiotic mainly works against Gram-positive bacteria, including mycobacteria, as well as Gram-negative bacteria, and yeast and mold are completely resistant to this antibiotic [181].

It is known that monensin A is a representative of a large group of natural polyether ionophore antibiotics and was discovered in 1967 by Agtarap and colleagues as a metabolite formed during the biosynthesis of *Streptomyces cinnamonensis* [182]. One of the monensin analogs is monensin C (**101**), which demonstrates activity against Gram-positive bacteria of the genera *Micrococcus, Bacillus*, and *Staphylococcus* [183,184].

The antibiotic named promomycin (**102**) and the related ionophoric polyethers A80438 (**103**), mutalomycin (**104**, 3D graph see in Figure 36), and lomonomycin (**105**), are produced by *Streptomyces scabrisporus* and have also been found in various *Streptomyces* spp., including *Streptomyces ribosidificus* and *S. mutabilis*. Promomycin and other antibiotics (**102**–**105**) found inhibit the growth of the Gram-positive bacteria *Bacillus subtilis* [185,186,187,188].

An antibiotic called octacyclomycin (**107**) was found in the fermentation broth of *Streptomyces* sp. No. 82–85 and showed both cytocidal activity against B16 melanoma cells and antimicrobial activity against Gram-positive bacteria in vitro [189].

An antibiotic named nanchangmycin (**108**, structure seen in Figure 37, predicted activity shown in Table 12, and 3D graph seen in Figure 38) was produced by *Streptomyces nanchangensis* NS3226 [143]. It demonstrated cytotoxic activity against several cancer cell lines, being most active against HL-60 (human leukemia) and HCT-116 (human colon carcinoma) cell lines, presenting IC_50_ and (IS) values: 0.0014 μM (30.0) and 0.0138 μM (3.0), respectively. On HCT-116, nigericin caused apoptosis and autophagy. Nigericin also showed activity against a panel of cancer-related kinases and inhibited both JAK3 and GSK-3β kinases in vitro and its binding affinities [190].

*Streptomyces scabrisporus* NF3, an endophytic actinomycete, which was isolated from *Amphipterygium adstringens* in Mexico, exhibited the potential to produce diverse bioactive compounds, for instance, the antibacterial hitachimycin and the antitumoral alborixin (**109**) [191]. 

A polyether antibiotic marked as 6016 (**110**) was isolated from the culture of *Streptomyces albus* strain No. 6016. The antibiotic exhibited activity against Gram-positive bacteria including mycobacteria and was effective in the treatment of coccidiosis of fowl [192,193]. A polyether ionophore antibiotic named cationomycin (**111**) was isolated from extracts of *Actinomadura* NOV sp. [194], and endusamycin (**112**) was isolated from *Streptomyces endus* [195].

An antibiotic named mutalomycin (**113**) is a new metal-complexing polyether antibiotic produced by a strain of *Streptomyces mutabilis* NRRL 8088. Mutalomycin contains six heterocyclic rings and is structurally related to nigericin. The metabolite is active against Gram-positive bacteria and *Eimeria tenella* (chicken coccidiosis) [186]. Lasalocid metal (**114**) was isolated from *Streptomyces lasaliensis*. Crystal structures of lasalocid acid barium, silver, and strontium salts were determined. The monomeric unit of lasalocid thallium salt is stabilized by strong, intramolecular aryl-Tl type-metal half sandwich bonding interactions. Homologs of lasalocid acid were also described [196,197,198].

Antibiotics 27C6 (**115**, for structure see Figure 39, predicted activity shown in Table 12) and K-41 (**116**), carboxylic polyether compounds, were isolated from *Leclercia adecarboxylata*, the strain KP-27C6, and *Streptomyces hygroscopicus* K41, respectively. Both molecules exhibited antibacterial activity against Gram-positive bacteria, anti-coccidal activity, and delayed toxicity for poultry in vivo [199]. The structure of CP-96,797 (**117**), a polyether antibiotic, is related to K-41A and produced by *Streptomyces* sp. [200].

An antibiotic named octacyclomycin (**118**) was found in the culture broth of *Streptomyces* sp. No. 82-85 [201] and is also produced by *Actinoallomurus* sp. ID14582 [202]. An antibiotic marked W341C (**119**) is a monocarboxylic polyether metabolite with anti-coccidal properties produced by *Streptomyces* sp. W341, and it demonstrated the ability of W341C to induce potassium loss in *Bacillus subtilis* and *Streptococcus agalactiae* and promote potassium uptake into *Escherichia coli* [203].

The culture broth of an isolate, *Streptomyces* sp. CS684, showed antibacterial activity on methicilin-resistant *Staphylococcus aureus* (MRSA) and vancomycin-resistant *Enterococci* (VRE). Among purified substances from the organism, CSU-1, which is active against MRSA and VRE, is identified as laidlomycin (**120**) [204]. 

A polyether antibiotic labeled CP-84,657 (**121**, structure seen in Figure 40 and activity prediction shown in Table 13) was isolated by solvent extraction from the fermentation broth of *Actinomadura* sp. (ATCC 53708). This antibiotic is among the most potent anti-coccidal agents known, effectively controlling the Eimeria species that are the major causative agents of chicken coccidiosis at doses of 5 Mg/kg or less in feed. It is also active in vitro against certain Gram-positive bacteria, as well as the spirochete, *Treponema hyodysenteriae* [205]. Another antibiotic ionophore, CP-54,883 (**122**), the molecule of which contains a polyether ring network and side chain terminated by an aromatic ring containing a phenoxy and two chlorine substituents, was found in the fermentation broth of *Actinomadura routienii* [206], and an antibiotic ionophore CP-80219 (**123**) was found in the fermentation broth of *Streptomyces hygroscopicus* ATCC 53626 [207].

Grisorixin (**124**) is an ionophorous antibiotic of the nigericin group isolated from cultures of a strain of *Streptomyces griseus*. It shows activity against Gram-positive bacteria and fungi but is also very toxic [208].

An antibiotic labeled SF-2487 (**125**) was isolated from a culture broth of *Actinomadura* sp. SF2487. It showed moderate activity against Gram-positive bacteria, but no activity against Gram-negative bacteria. SF-2487 exhibited in vitro antiviral activity against influenza virus [209]. Another antibiotic labeled X-14931A (**126**) was isolated from a culture of *Streptomyces* sp. X-14931. Antibiotic X-14931A showed in vitro activity against Gram-positive microorganisms and yeasts. It was also active against mixed Eimeria infection in chickens and exhibited activity in the rumen growth [210].

Two polyether ionophores, X-14873A (**127**, structure seen in Figure 41, predicted activity shown in Table 14) and X-14873H (**128**, 3D graph seen in Figure 42) were isolated from the fermentation of *Streptomyces* sp. X-14873 (ATCC31679). Antibiotic X-14873A was mainly active against Gram-positive bacteria, and X-14873H, the descarboxyl derivative of X-14873A, was also active against Gram-positive bacteria [211].

An antibiotic named noboritomycin (**129**) was isolated from *Streptomyces noboritoensis*. It was a polyether ionophore possessing two carboxylic acid functions on the carbon backbone, namely a free acid and an additional carboxylic acid ethyl ester group. This antibiotic was active against a wide range of Gram-positive bacteria [212]. The antibiotic 6-chloronoboritomycin (**130**) was isolated from *S. malachitofuscus*. It was active against Gram-positive bacteria and some anaerobes. In addition, it exhibited in vitro activity against several strains of *Treponema hyodysenteriae*, a causal agent of swine dysentery [213].

Antibiotic CP-82009 (**131**) was isolated by solvent extraction from the fermentation broth of *Actinomadura* species (specimen of this species seen in Figure 43). It exhibited activity against Gram-positive bacteria, as well as the spirochete *Treponema hyodysenteriae* [214]. An antibiotic named abierixin (**132**) was isolated from *Streptomyces albus*. It exhibited weak activity against Gram-positive bacteria [215].

A potent polyether ionophore antibiotic named maduramicin (**133**, structure seen in Figure 44, and activity shown in Table 14) produced by *Actinomadura rubra* and *Actinomadura yumaensis* NRRL12515 showed anthelmintic properties and is an antiprotozoal agent used in veterinary medicine to prevent coccidiosis [216,217]. 

An interesting antibiotic called lenoremycin (**134**) that demonstrated antimicrobial activity (up to 60 μM), cancer cell line cytotoxicity (up to 20 μM), and displayed antibacterial and antifungal activities, is produced by *Streptomyces* sp. RM-14-6 [218].

Carriomycin (**135**), a polyether antibiotic, was isolated from the culture broth of *Streptomyces irygroscoprcus* T-42082. It is active against Gram-positive bacteria, several fungi, yeasts, and mycoplasma [219,220]. SY-4 (**136**) is 5-hydroxysalinomycin, produced by a strain of *Streptomyces albus* (ATCC 21838) [158].

A polyether antibiotic, kijimicin (**137**, 3D graph seen in Figure 45) was found in the culture nitrate of *Actinomadura* sp. MI215-NF3, which was isolated from a soil sample collected at Bunkyo-ku, Tokyo, Japan. The antibiotic showed higher anticoccidial activity than monensin or salinomycin [221,222].

Actinomycete *Actinomadura* sp. produced the antibiotic cationomycin (**138**) and its biosynthesis has been studied [223,224]. 

Ferensimycins A (**139**) and B (**140**) were isolated, as were their sodium salts, from the fermentation broth of *Streptomyces* sp. No. 5057. Both antibiotics are active against Gram-positive bacteria [225]. Semduramicin (**141**, 3D graph seen in Figure 45) was isolated from *Actinomadura roserufa* [226], and this antibiotic showed activity against five species of poultry Eimeria [227].

## 5. Structure–Activity Relationships and Biological Activities of Natural Polyether Ionophores 

It is known that the chemical structure of both natural and synthetic molecules predetermines biological activity, which makes it possible to analyze the structure–activity relationships (SARs). Such a wise idea was first proposed by Brown and Fraser more than 150 years ago, in 1868 [228]; although, according to other sources, SAR originates from the field of toxicology, according to which Cros, in 1863, determined the relationship between the toxicity of primary aliphatic alcohols and their solubility in water [229]. More than 30 years later, Richet in 1893 [230], Meyer in 1899 [231], and Overton in 1901 [232] separately found a linear correlation between lipophilicity and biological effects. By 1935, Hammett [233,234] presented a method of accounting for the effect of substituents on reaction mechanisms using an equation that considered two parameters, namely the substituent constant and the reaction constant. Complementing Hammett’s model, Taft proposed, in 1956, an approach for separating the polar, steric, and resonance effects of substituents in aliphatic compounds [235]. Combining all previous developments, Hansch and Fujita laid out the mechanistic basis for the development of the QSAR method [236], and the linear Hansch equation and Hammett’s electronic constants are detailed in the book by Hansch and Leo published in 1995 [237]. 

Some well-known computer programs can, with some degree of reliability, estimate the pharmacological activity of organic molecules isolated from natural sources or synthesized compounds [238,239,240]. It is known that classical SAR methods are based on the analysis of (quantitative) structure–activity relationships for one or more biological activities, using organic compounds belonging to the same chemical series as the training set [241]. The computer program PASS, which has been continuously updating and improving for the past thirty years [20], is based on the analysis of a heterogeneous training set including information about more than 1.3 million known biologically active compounds with data on ca. 10,000 biological activities [20,22]. Chemical descriptors implemented in PASS, which reflect the peculiarities of ligand–target interactions and the original realization of the Bayesian approach for elucidation of structure–activity relationships, provide the average accuracy and predictivity for several thousand biological activities equal to about 96% [20,21]. In several comparative studies, it was shown that PASS outperforms, in predictivity, some other recently developed methods for the estimation of biological activity profiles [20,21,22]. Freely available via the Internet, the PASS Online web service [242] is used by more than thirty thousand researchers from almost a hundred countries to determine the most promising biological activities for both natural and synthetic compounds [243,244,245,246,247]. To reveal the hidden pharmacological potential of the natural substances, we have successfully used PASS for the past fifteen years [248,249,250,251,252,253,254].

## 6. Conclusions

In connection with the intensive development of information technology in the last decade, a lot of computer programs have appeared that use algorithms to determine the biological activity of various natural and synthetic molecules. Many of them, as a rule, are not effective enough for these purposes. However, there are programs that have proven their effectiveness for several decades. The data on the biological activity of natural polyether ionophores given in this review were obtained both experimentally and using PASS. The data comparison shows that natural polyether ionophores exhibit strong antitumor, antifungal, antibacterial, and antimicrobial activities, among many other activities. The presented data may be of interest to specialists in various fields of science such as organic chemistry and molecular biology; however, these data are of the greatest interest in a practical use for pharmacologists, physicians, and in applied medicine.

## Figures and Tables

**Figure 1 marinedrugs-20-00292-f001:**
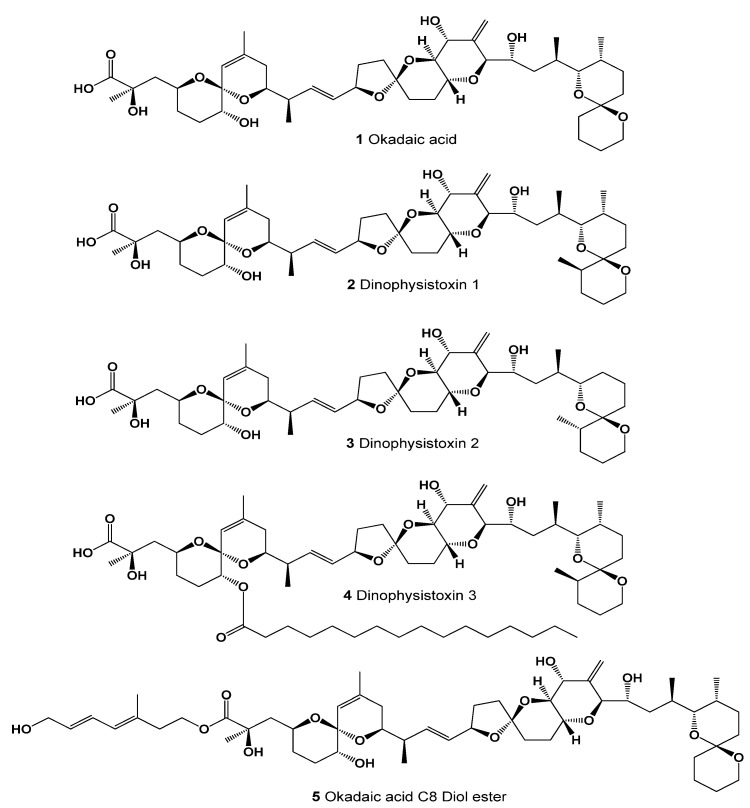
Okadaic acid, dinophysistoxins, and derivatives.

**Figure 2 marinedrugs-20-00292-f002:**
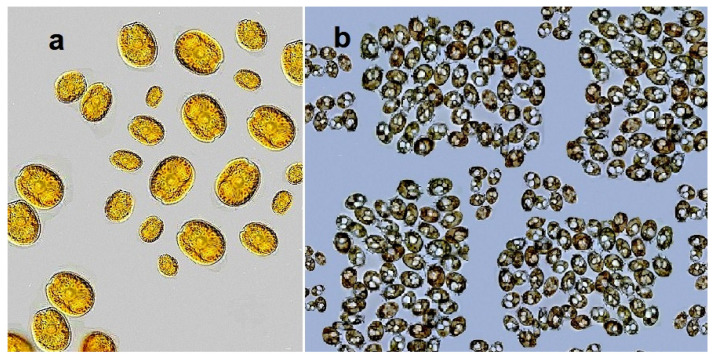
Samples of the dinoflagellates of the genera *Prorocentrum* (**a**) and *Dinophysis* (**b**), which synthesize the best-known marine polyether toxin called okadaic acid. All photos are taken from sites where permission is granted for non-commercial use.

**Figure 3 marinedrugs-20-00292-f003:**
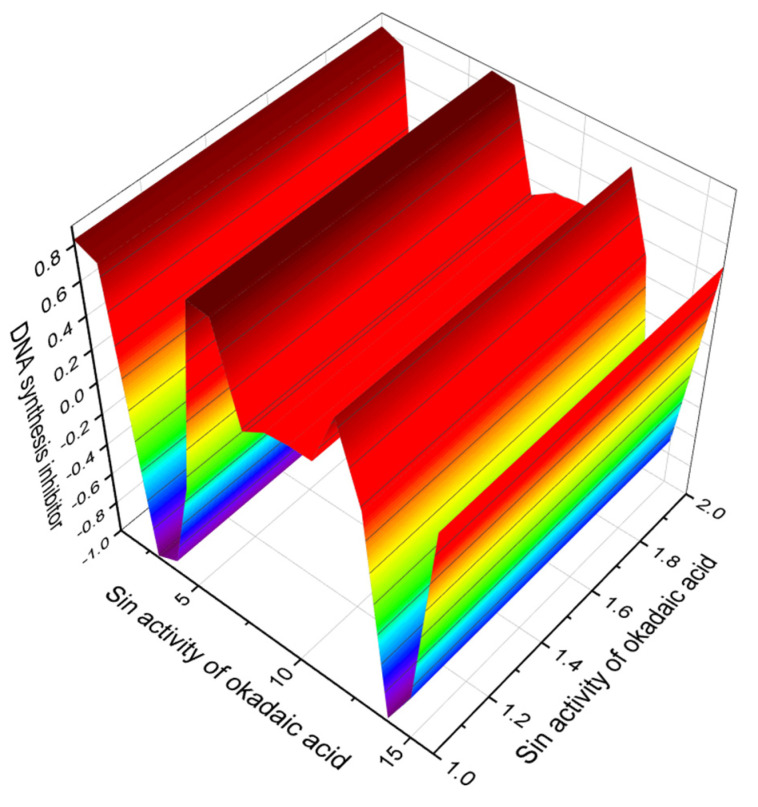
3D graph showing the predicted and calculated biological property of okadaic acid (**1**) as an inhibitor of DNA synthesis with the highest degree of confidence (99.1%).

**Figure 4 marinedrugs-20-00292-f004:**
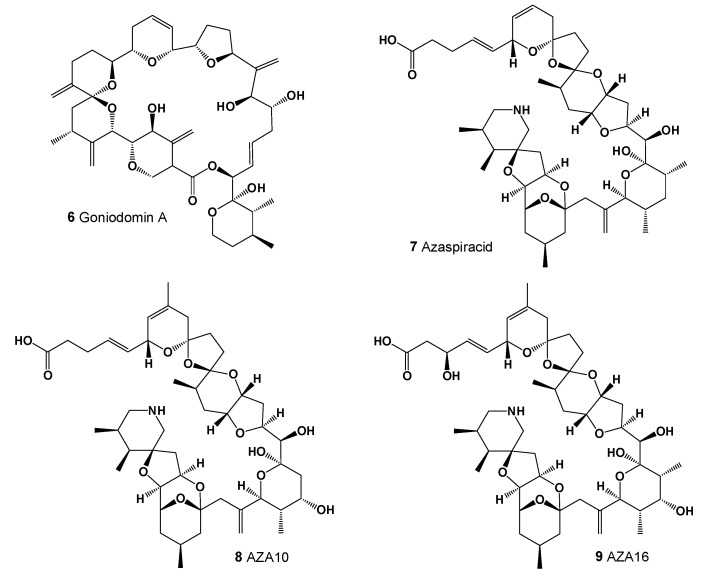
Lipophilic polyether toxins.

**Figure 5 marinedrugs-20-00292-f005:**
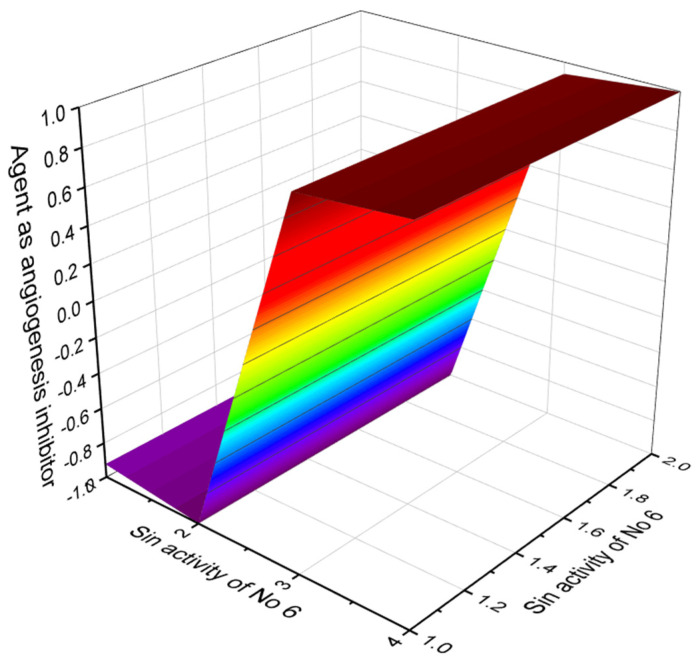
3D graph showing the predicted and calculated biological property of goniodomin A (**6**) as angiogenesis inhibitor with the highest degree of confidence (97.9%).

**Figure 6 marinedrugs-20-00292-f006:**
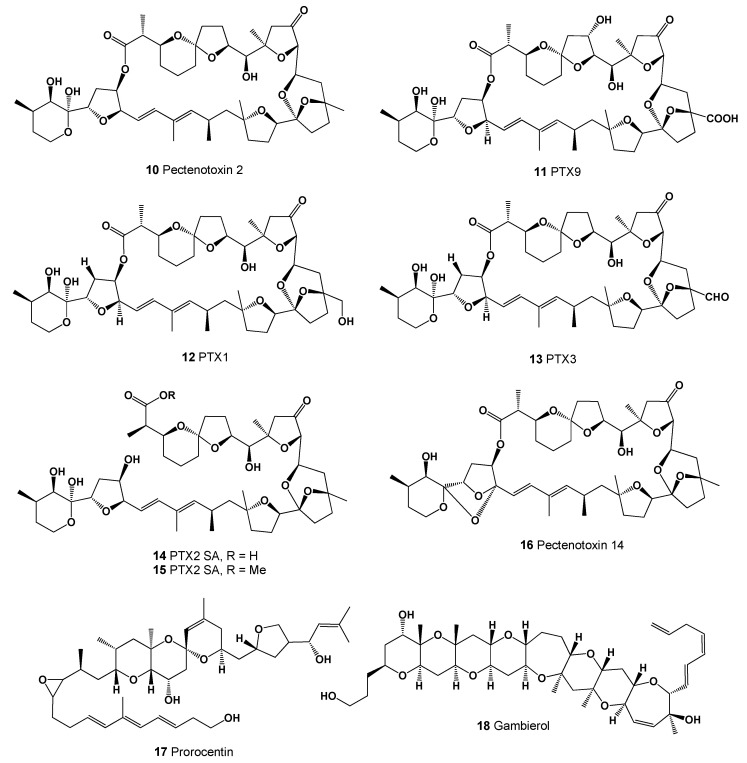
Marine polyether pectenotoxins, and prorocentin and gambierol.

**Figure 7 marinedrugs-20-00292-f007:**
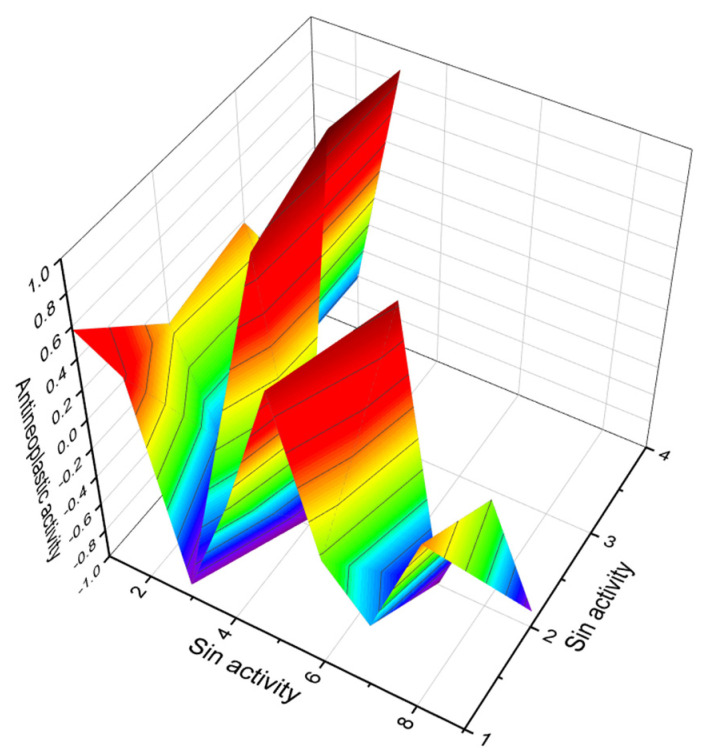
3D graph showing the predicted and calculated antineoplastic activity of polyether pectenotoxins (compound numbers: **10**, **11**, **14**, and **15**) showing the highest degree of confidence, more than 93%.

**Figure 8 marinedrugs-20-00292-f008:**
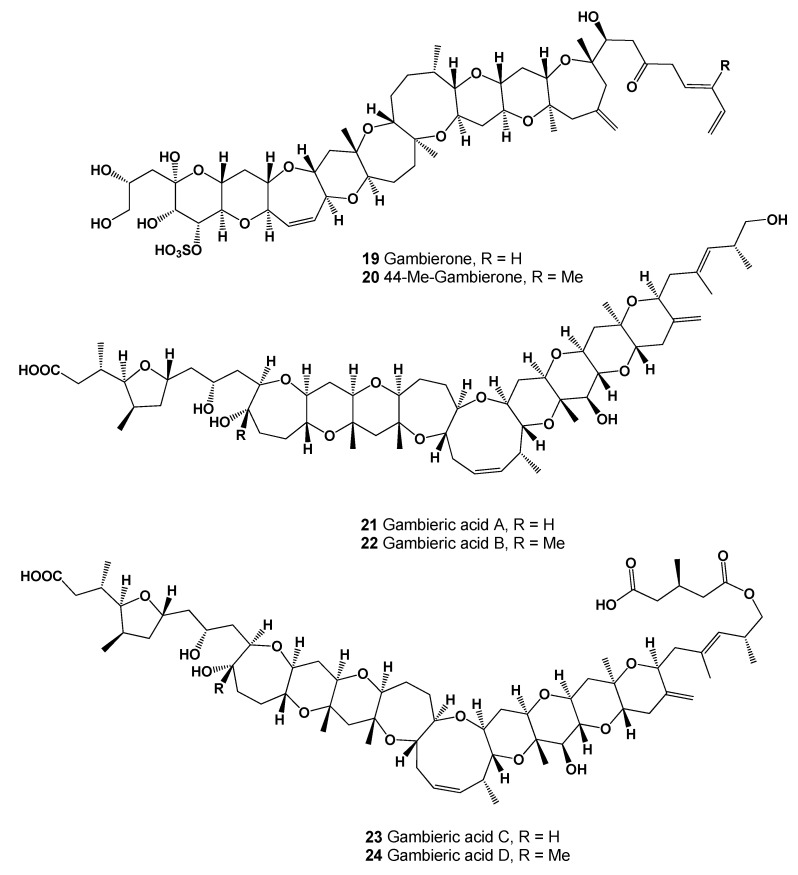
Polycyclic ether toxins are produced by the benthic dinoflagellate of the genus Gambierdiscus (**19**–**24**).

**Figure 9 marinedrugs-20-00292-f009:**
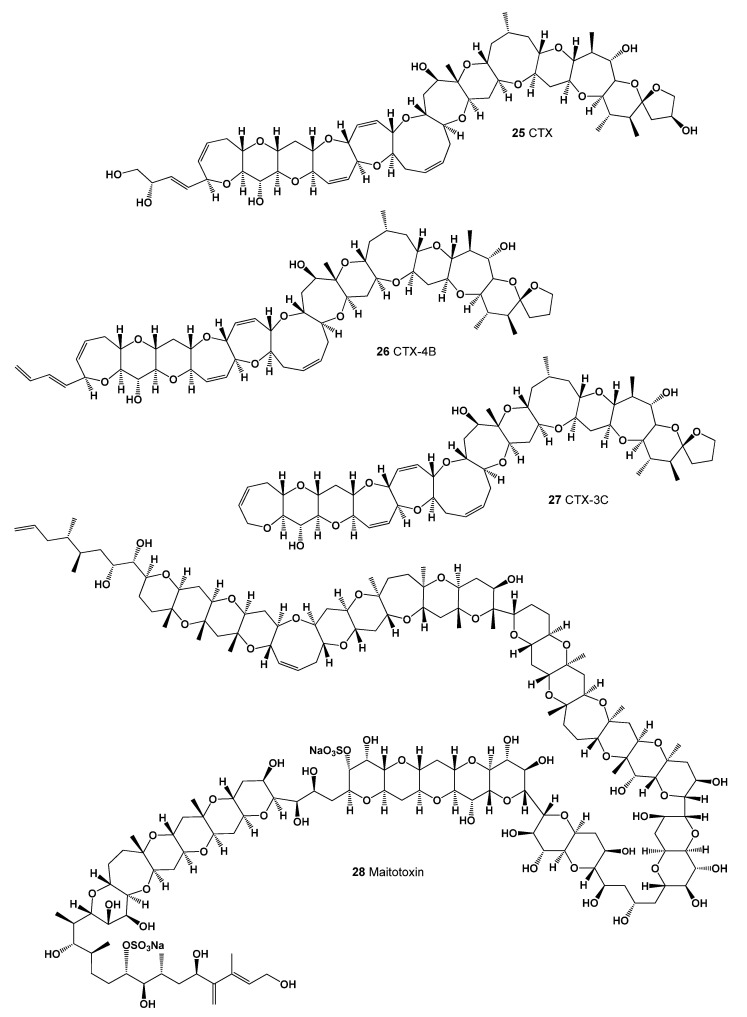
Polycyclic ether toxins are produced by the benthic dinoflagellate of the genus Gambierdiscus (**25**–**28**).

**Figure 10 marinedrugs-20-00292-f010:**
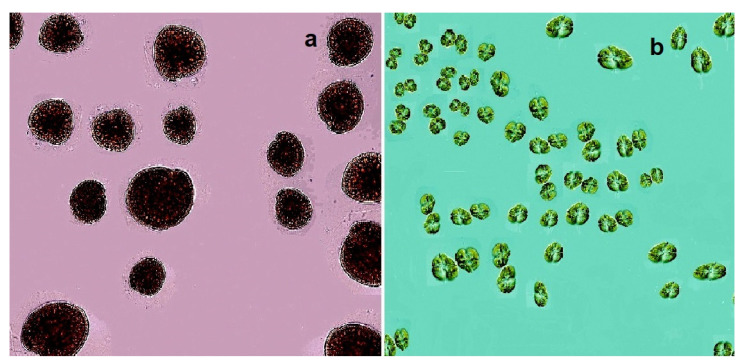
Samples of the dinoflagellates that produce red tides in various parts of the world’s oceans are (**a**) *Gambierdiscus toxicus* and (**b**) *Karenia brevis*. These microalgae scavenge ciguatoxins and brevetoxins and other polyether metabolites.

**Figure 11 marinedrugs-20-00292-f011:**
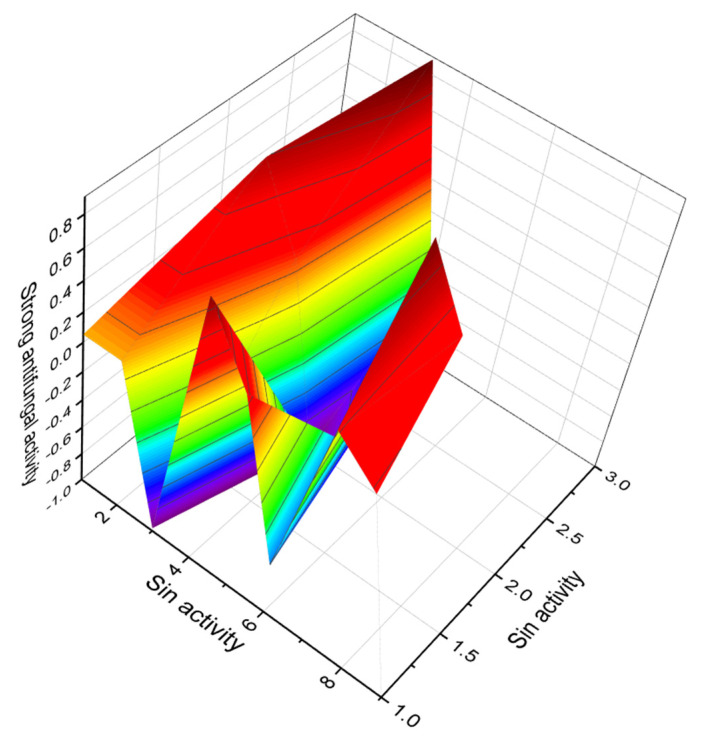
3D graph showing the predicted and calculated strong antifungal activity of polyether compounds (compound numbers: **23**, **24**, and **28**) showing the highest degree of confidence, more than 93%.

**Figure 12 marinedrugs-20-00292-f012:**
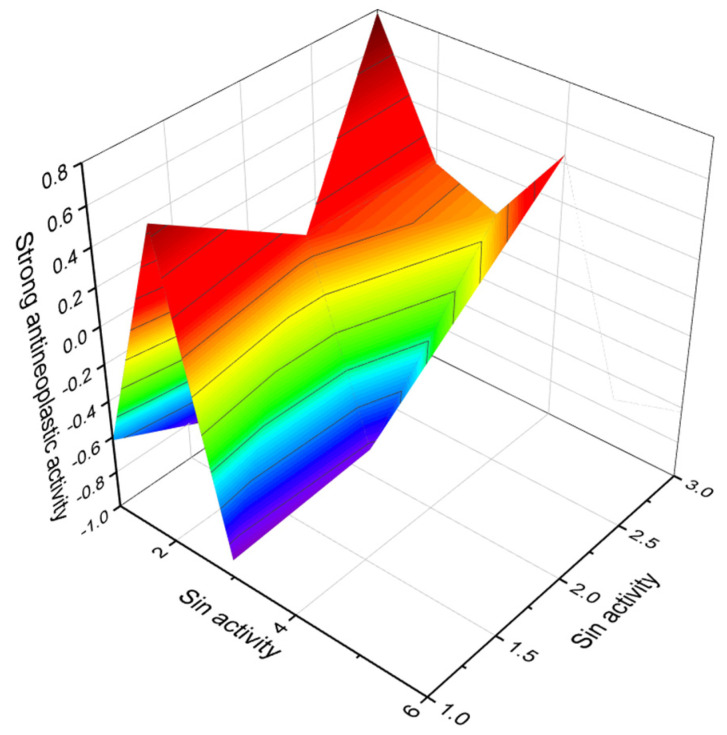
3D graph showing the predicted and calculated strong antineoplastic activity of polyether compounds (compound numbers: **21**, **25**, and **27**) showing the highest degree of confidence, more than 94%.

**Figure 13 marinedrugs-20-00292-f013:**
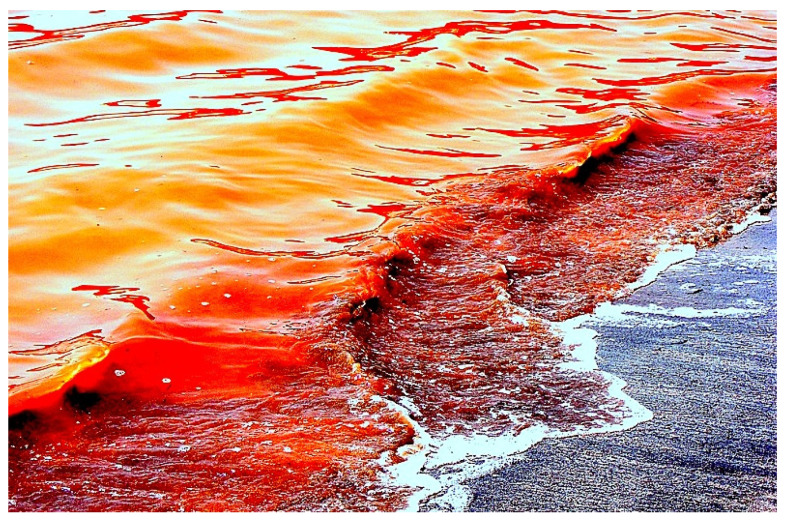
The red tide in Florida is a high concentration of naturally occurring marine dinoflagellates called *Karenia brevis*. These microalgae produce brevetoxins, powerful neurotoxins that can kill marine invertebrates, fish, animals, macroalgae, and can also be dangerous to humans.

**Figure 14 marinedrugs-20-00292-f014:**
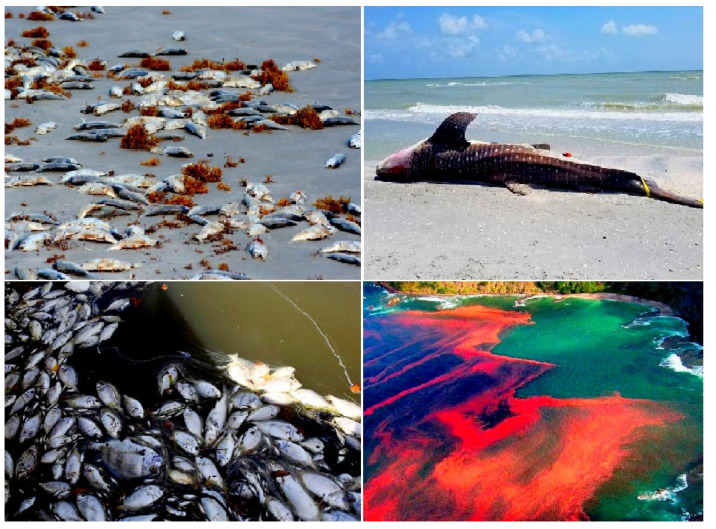
The red tide in Florida caused catastrophic consequences, and above all, it was the mass death of marine fish, animals, and marine invertebrates. The number of dead fish and animals is estimated at tens of thousands of tons. Red tides in Florida affect areas in southwest Florida including Palm Beach, Martin, St. Lucie, Glades, Hendry, Lee, and Okeechobee counties. The tourism business has been deeply affected because people do not want to be outdoors due to the strong smell and cough that the red tide causes. Fish kills and respiratory problems in humans can occur when microalgae cell levels reach 10,000 cells per liter, and cell concentrations as high as 2.5 million cells per liter have been found in some locations near the Sanibel Lighthouse.

**Figure 15 marinedrugs-20-00292-f015:**
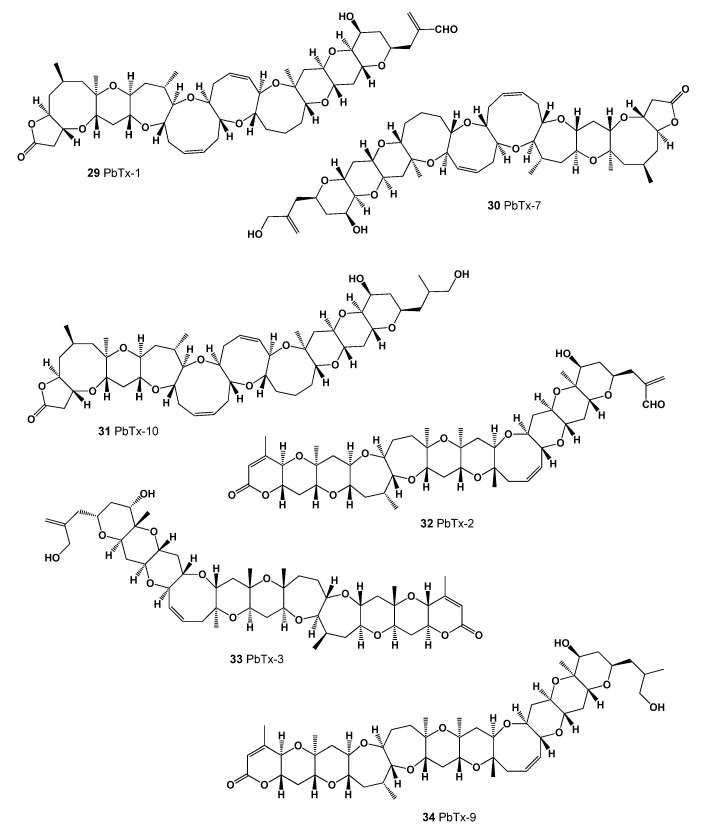
*Karenia brevis* produced a set of potent neurotoxins called brevetoxins.

**Figure 16 marinedrugs-20-00292-f016:**
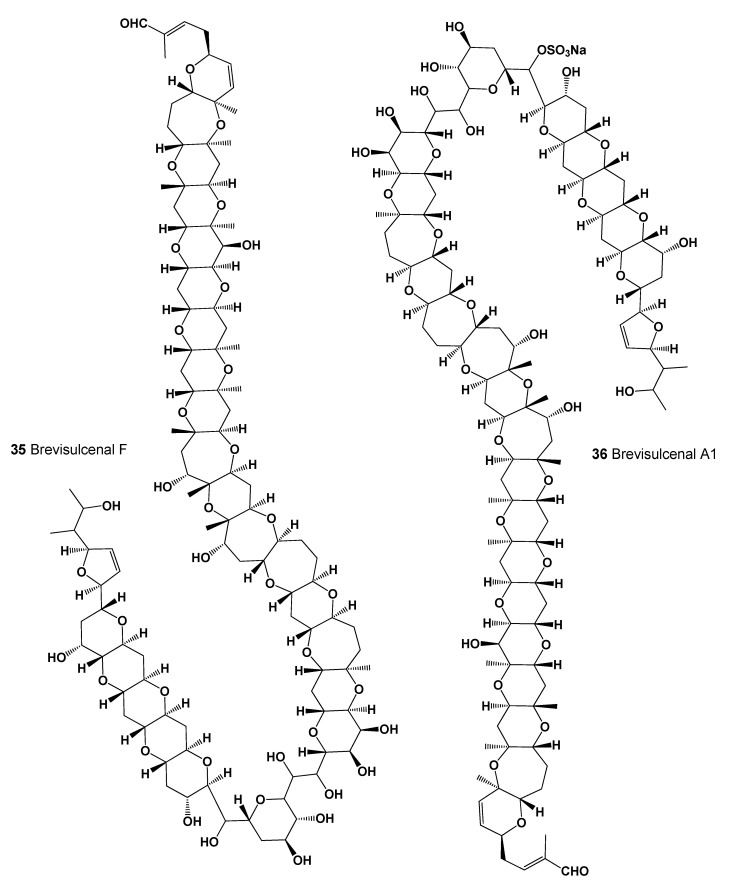
Two interesting polycyclic ether toxins have been found in the dinoflagellate *Karenia brevisulcata*.

**Figure 17 marinedrugs-20-00292-f017:**
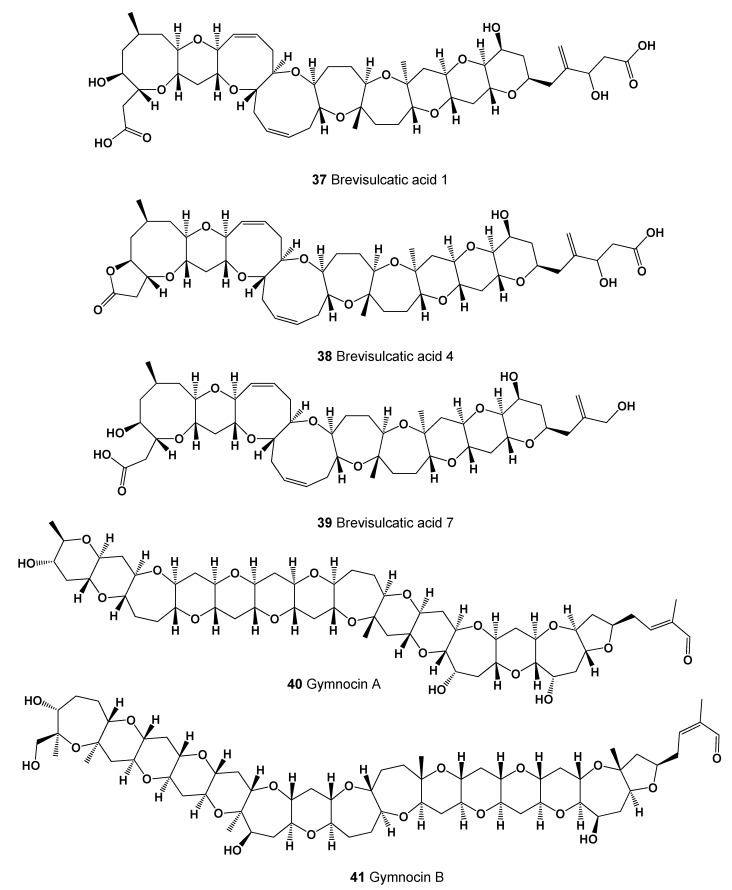
Brevisulcatic acids are produced by the red tide dinoflagellate *Karenia brevisulcata,* and gymnocin A and B are derived from the dinoflagellate, *Gymnodinium mikimotoi*.

**Figure 18 marinedrugs-20-00292-f018:**
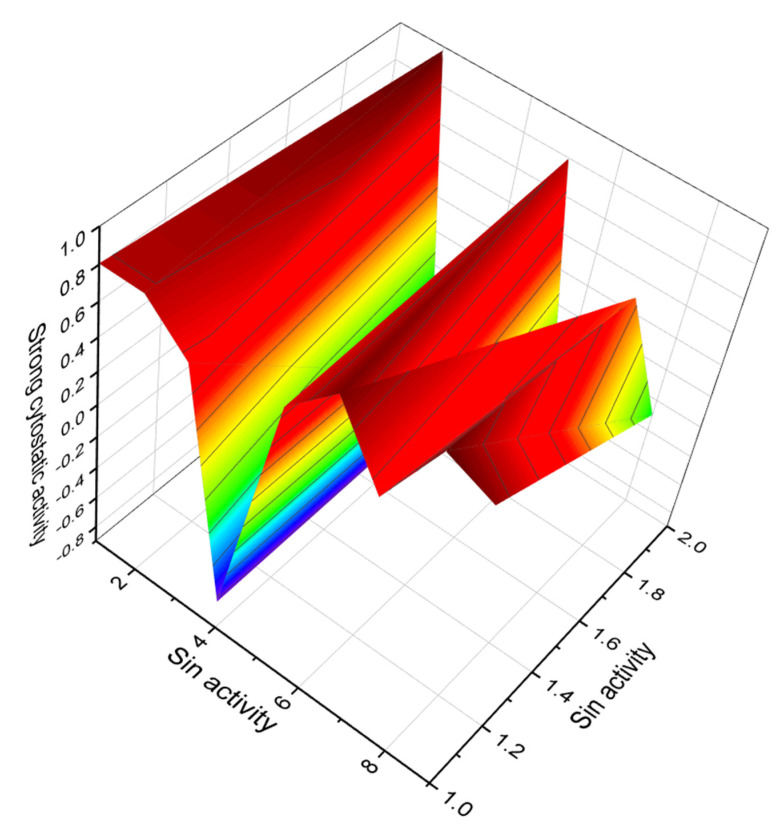
3D graph showing the predicted and calculated strong cytostatic activity of polyether compounds (compound numbers: **37** and **39**) showing the highest degree of confidence, more than 91%.

**Figure 19 marinedrugs-20-00292-f019:**
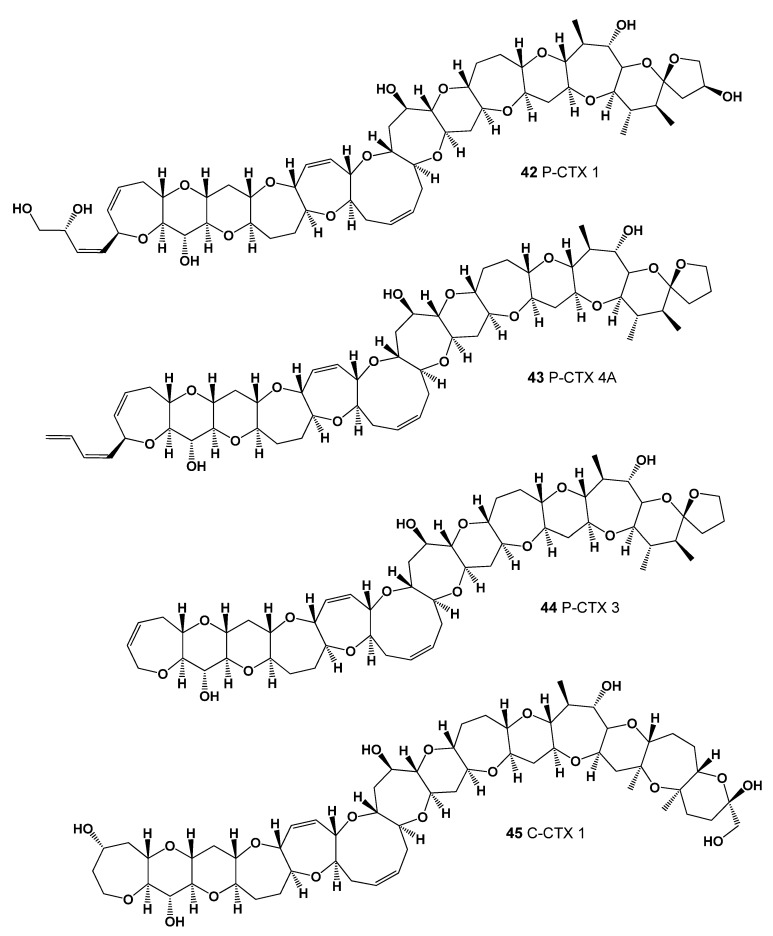
Ciguatoxins, polyether toxins derived from marine dinoflagellates.

**Figure 20 marinedrugs-20-00292-f020:**
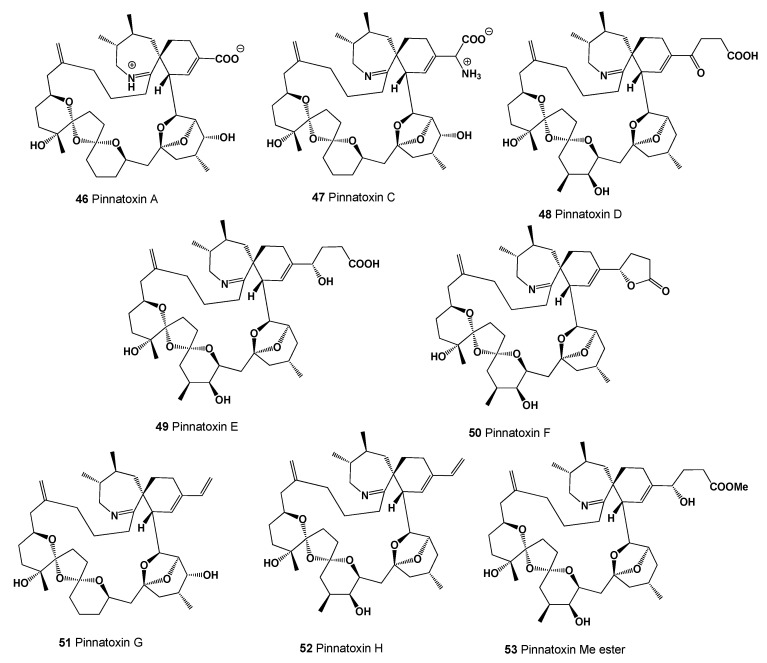
Polycyclic ether pinnatoxins were first found in the bivalve mollusc Pinna muricata.

**Figure 21 marinedrugs-20-00292-f021:**
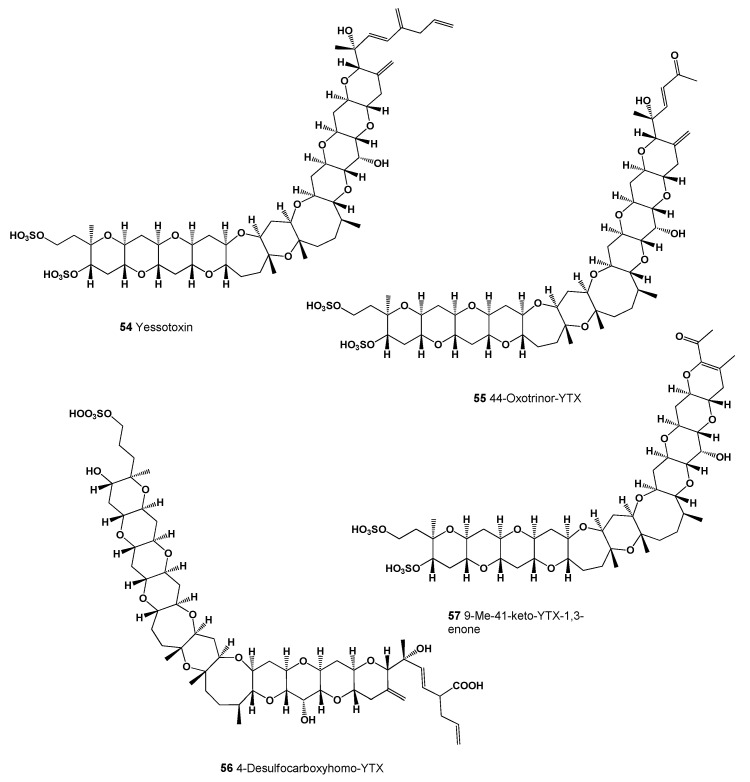
A disulfated polyether toxin, yessotoxin, is produced by marine dinoflagellates and accumulated in filter feeding shellfish.

**Figure 22 marinedrugs-20-00292-f022:**
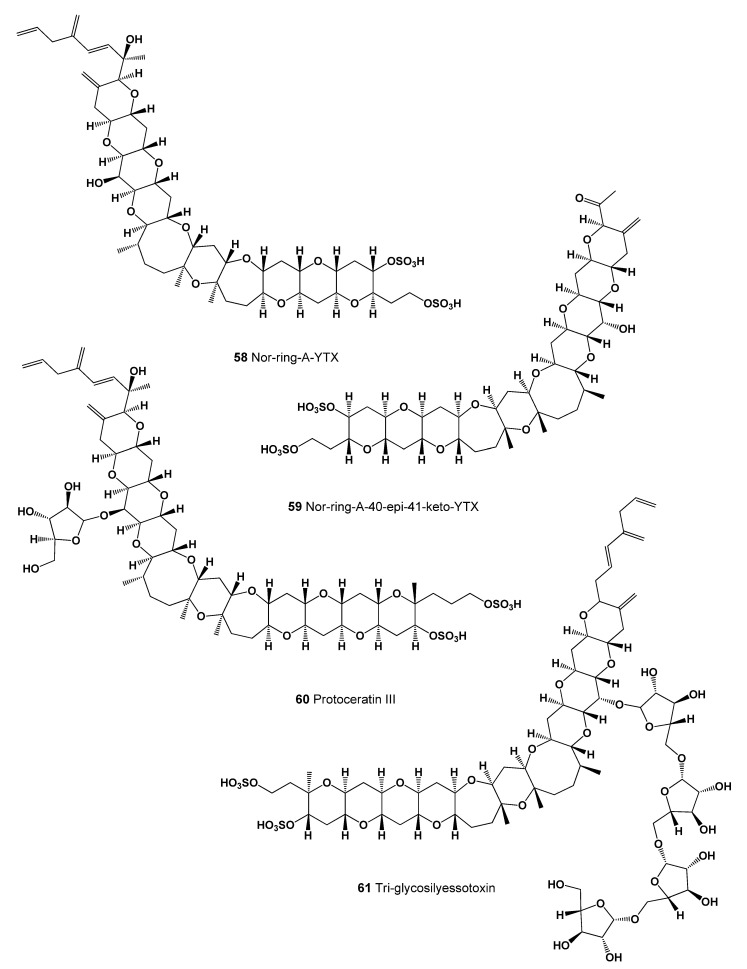
Yessotoxin analogues are produced by marine dinoflagellates and accumulated in filter feeding shellfish.

**Figure 23 marinedrugs-20-00292-f023:**
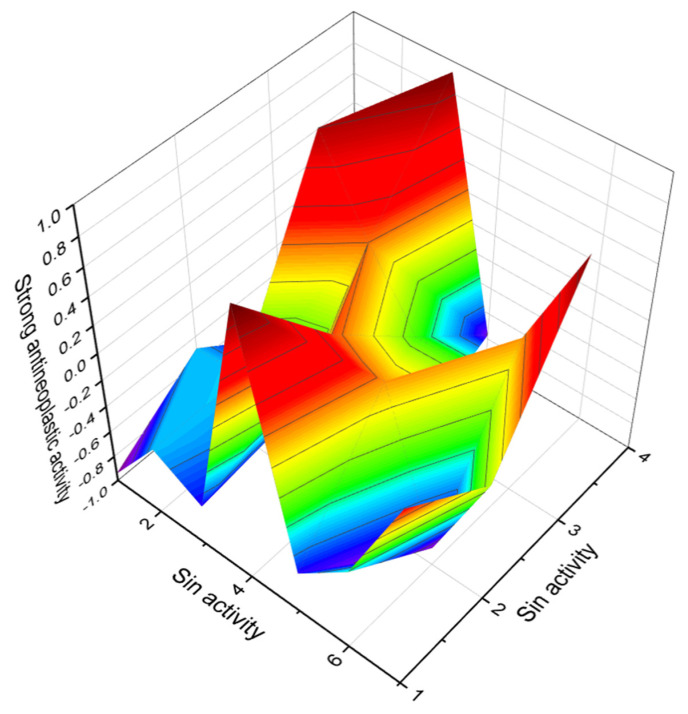
3D graph showing the predicted and calculated activity with strong antineoplastic (compound numbers: **54**, **58**, **60**) and antibacterial (compound number **61**) properties of polyether compounds with the highest degree of confidence, more than 91%.

**Figure 24 marinedrugs-20-00292-f024:**
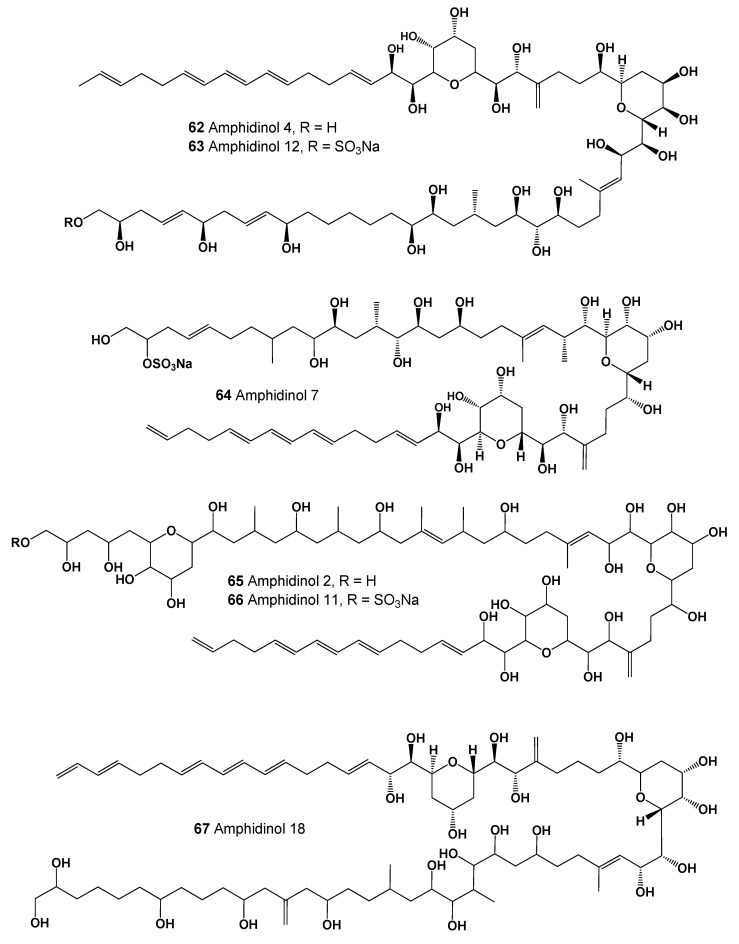
A class of amphidinol-like compounds named karlotoxins produced by mixotrophic strains of the dinoflagellate *Karlodinium veneficum* and some *Amphidinium* species.

**Figure 25 marinedrugs-20-00292-f025:**
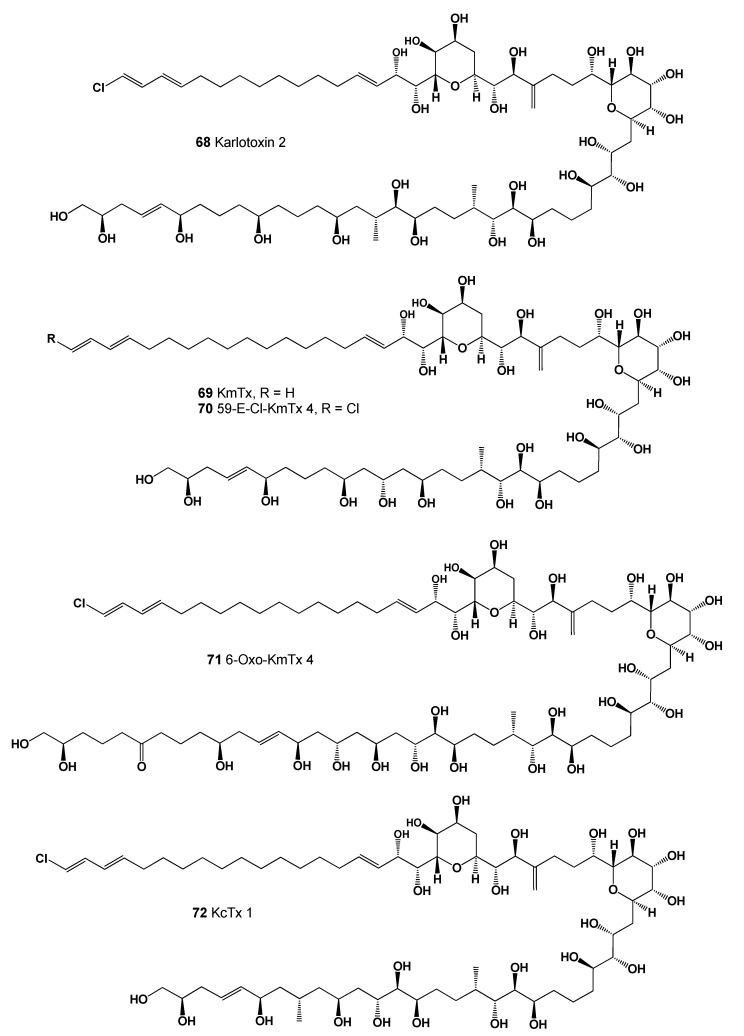
A class of amphidinol-like compounds named karlotoxins produced by mixotrophic strains of the dinoflagellate *Karlodinium veneficum* and several *Amphidinium* species.

**Figure 26 marinedrugs-20-00292-f026:**
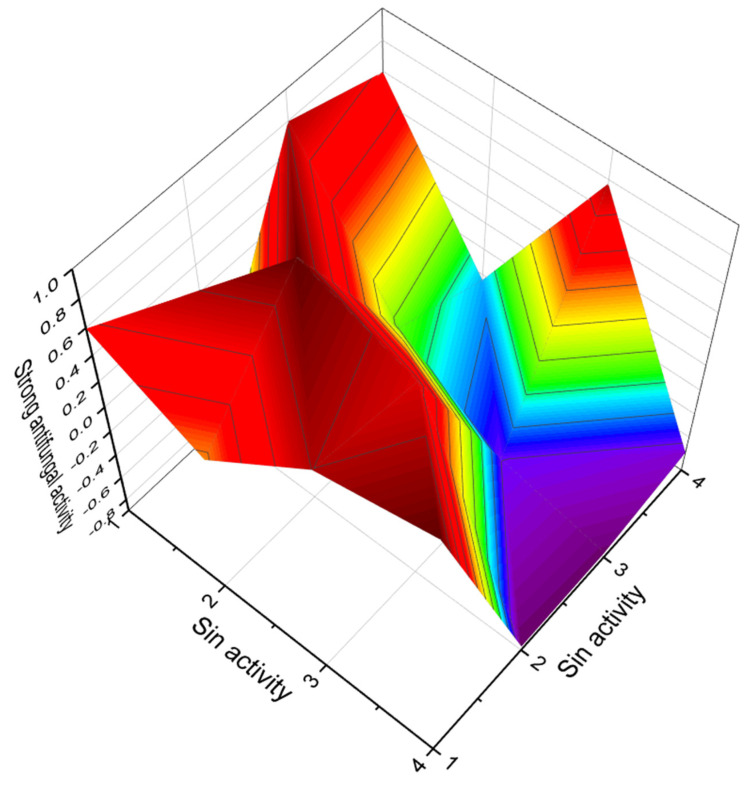
3D graph showing the predicted and calculated moderate antifungal activity of polyether compounds (compound numbers: **66**, **69**, **71**, and **72**) showing the highest degree of confidence, more than 90%.

**Figure 27 marinedrugs-20-00292-f027:**
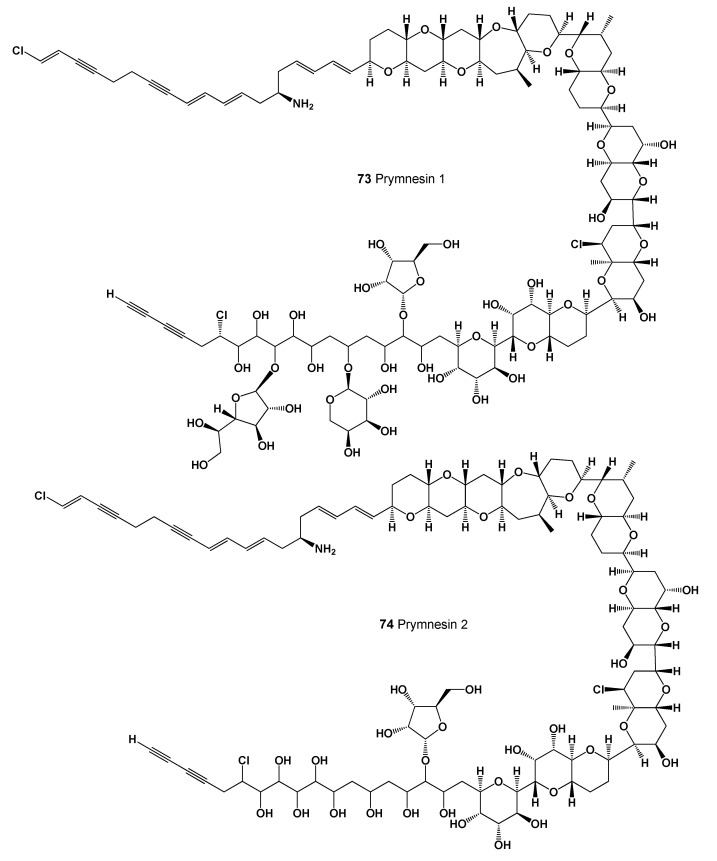
The golden alga *Prymnesium parvum* produces phycotoxins prymnesins.

**Figure 28 marinedrugs-20-00292-f028:**
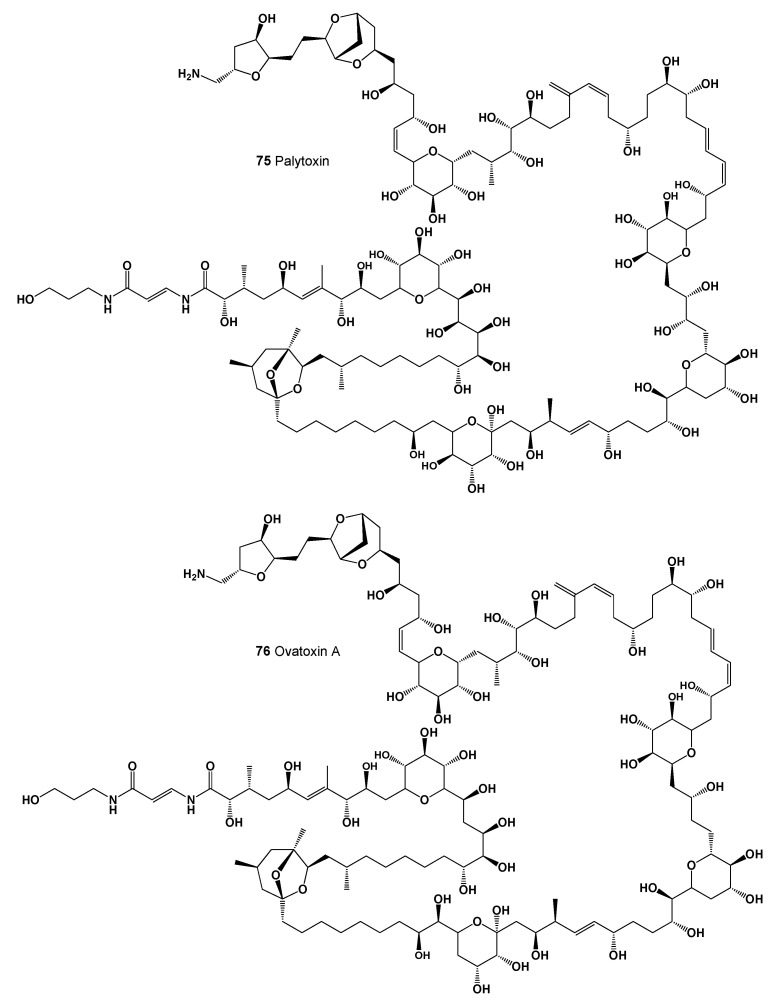
Marine lipophilic polyether toxins.

**Figure 29 marinedrugs-20-00292-f029:**
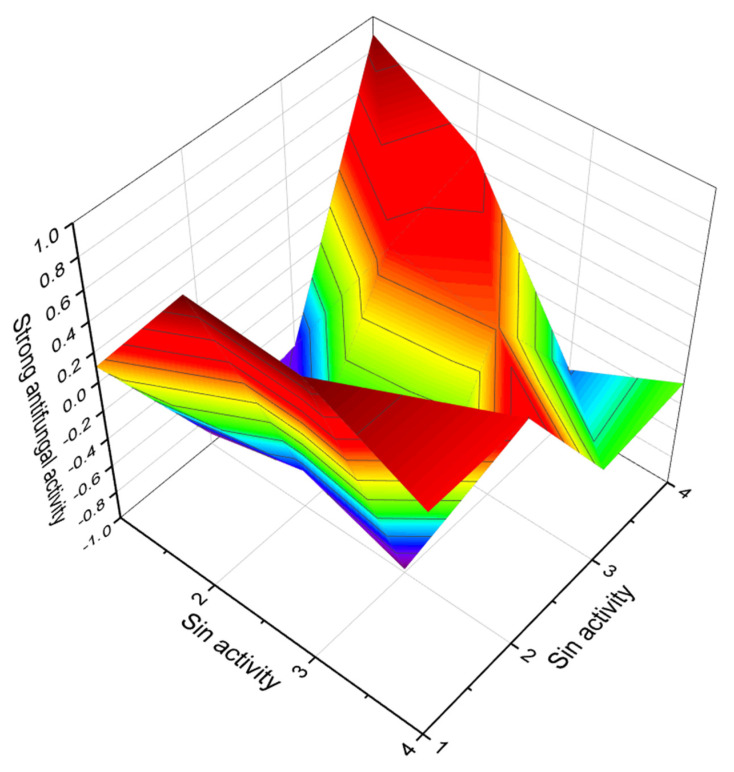
3D graph showing the predicted and calculated strong antifungal activity of polyether compounds (compound numbers: **73**, **74**, **75**, and **76**) with the highest degree of confidence, more than 97%.

**Figure 30 marinedrugs-20-00292-f030:**
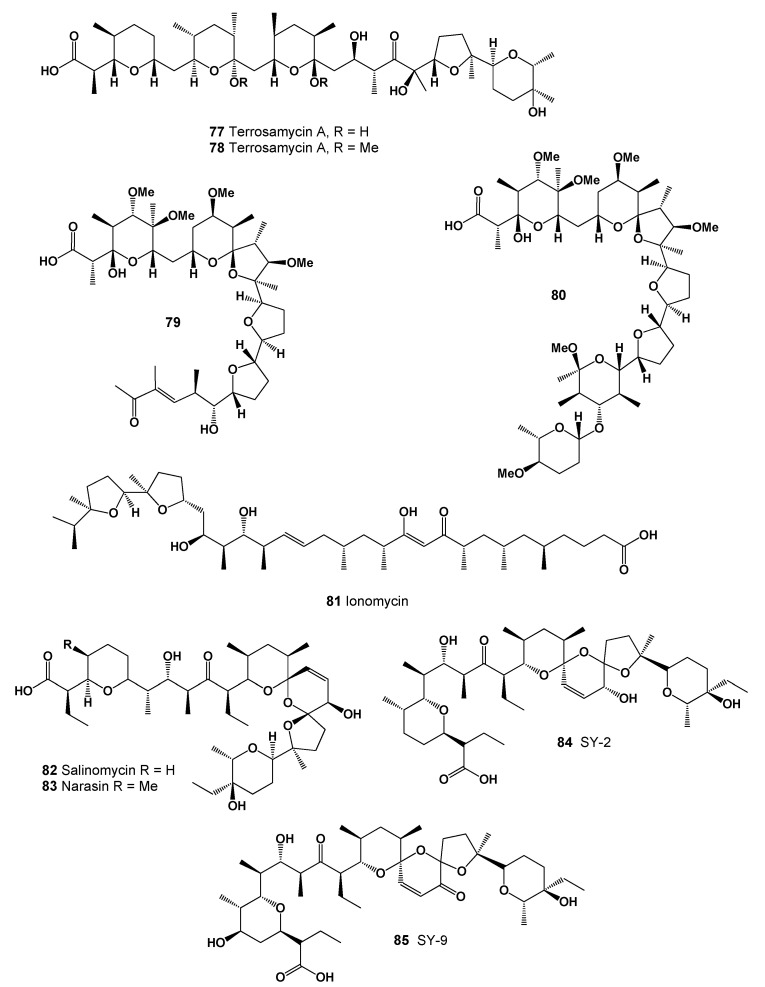
Ionophoric polyether antibiotics produced by Streptomycetes.

**Figure 31 marinedrugs-20-00292-f031:**
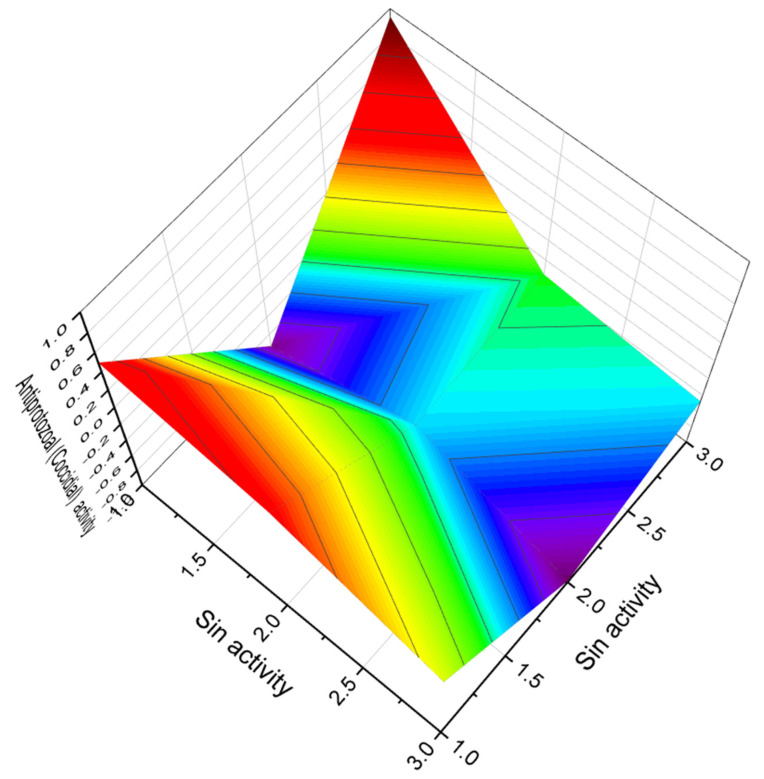
3D graph showing the predicted and calculated antiprotozoal (*Coccidial*) activity of polyether compounds (compound numbers: **77**, **78**, and **79**) with the highest degree of confidence, more than 90%.

**Figure 32 marinedrugs-20-00292-f032:**
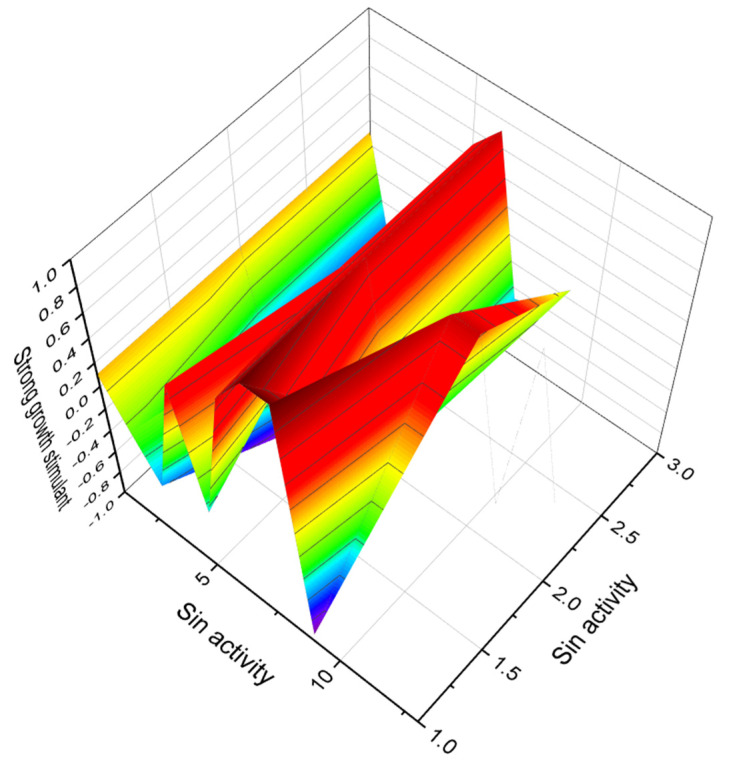
3D graph showing the predicted and calculated activity as growth stimulant of polyether compounds (compound numbers: **82**, **83**, and **84**) with the highest degree of confidence, more than 97%.

**Figure 33 marinedrugs-20-00292-f033:**
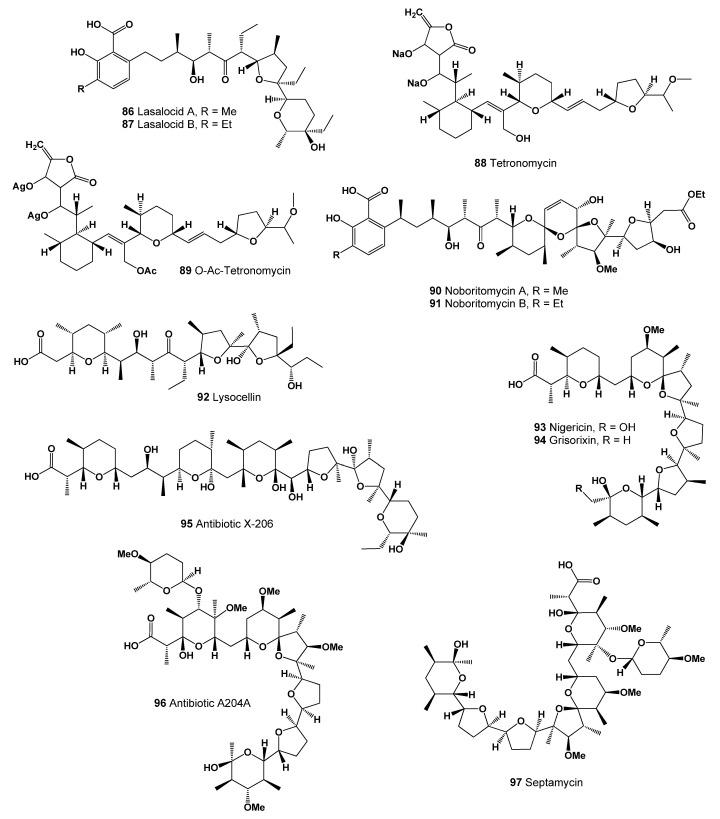
Ionophoric polyethers produced by Actinomycetes.

**Figure 34 marinedrugs-20-00292-f034:**
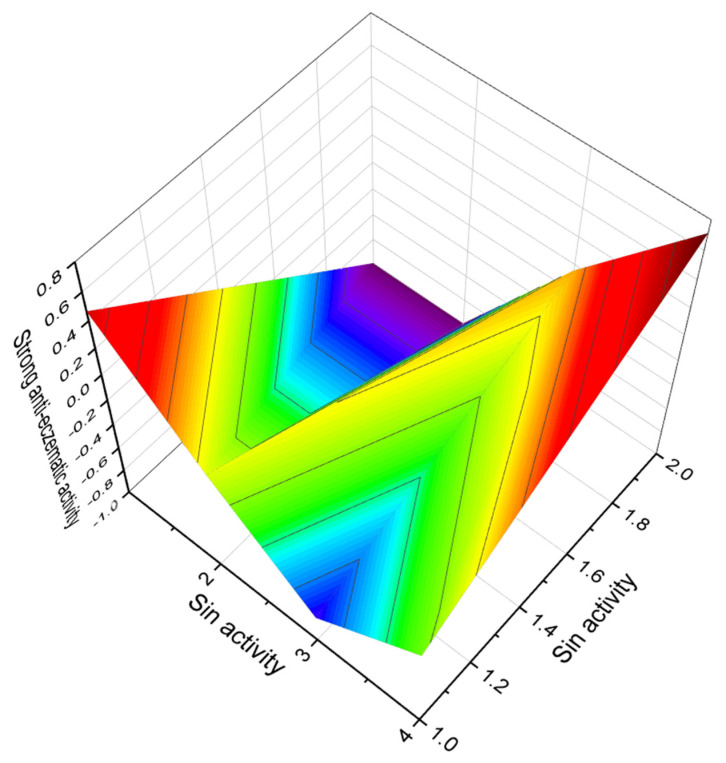
3D graph showing the predicted and calculated anti-eczematic activity of polyether compounds (compound numbers: **88** and **89**) showing the highest degree of confidence, more than 92%.

**Figure 35 marinedrugs-20-00292-f035:**
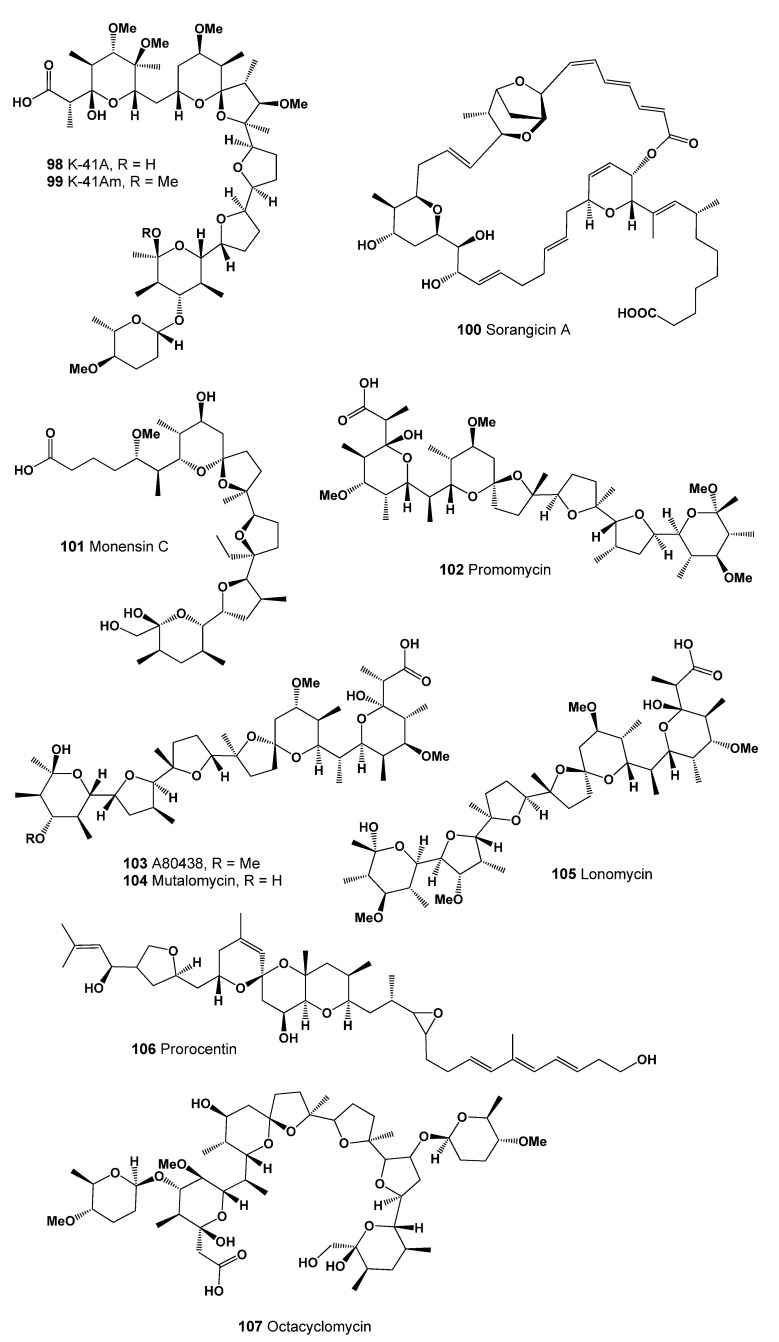
Bioactive polyether antibiotics produced by Actinomycetes.

**Figure 36 marinedrugs-20-00292-f036:**
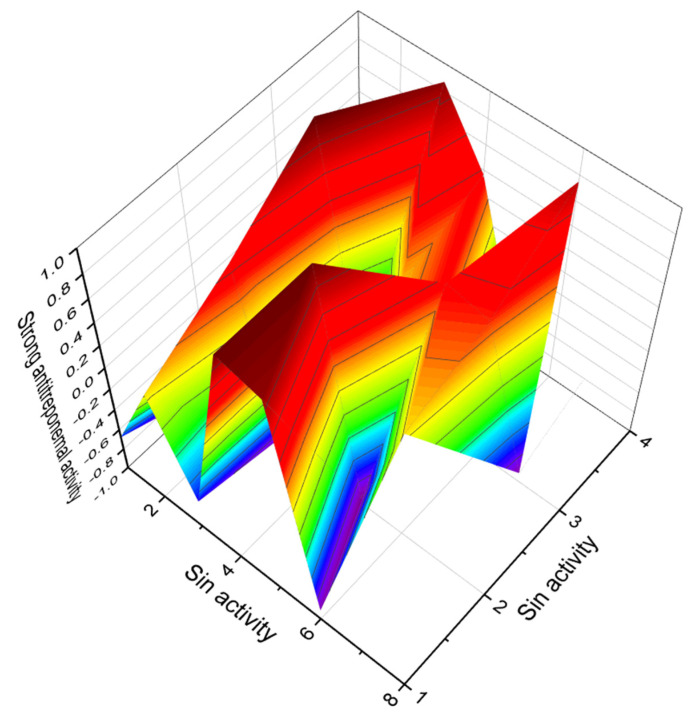
3D graph showing the predicted and calculated anti-treponemal activity of polyether compounds (compound numbers: **98**, **99**, **103**, and **104**) with the highest degree of confidence, more than 92%.

**Figure 37 marinedrugs-20-00292-f037:**
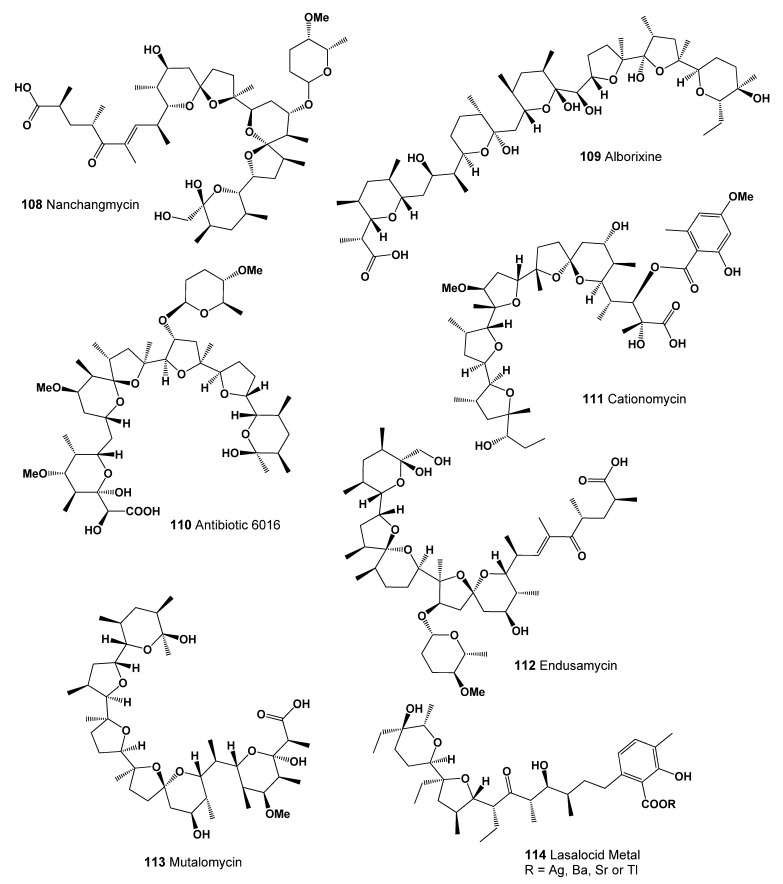
Lipophilic polyether antibiotics produced by Actinomycetes (**108**–**114**).

**Figure 38 marinedrugs-20-00292-f038:**
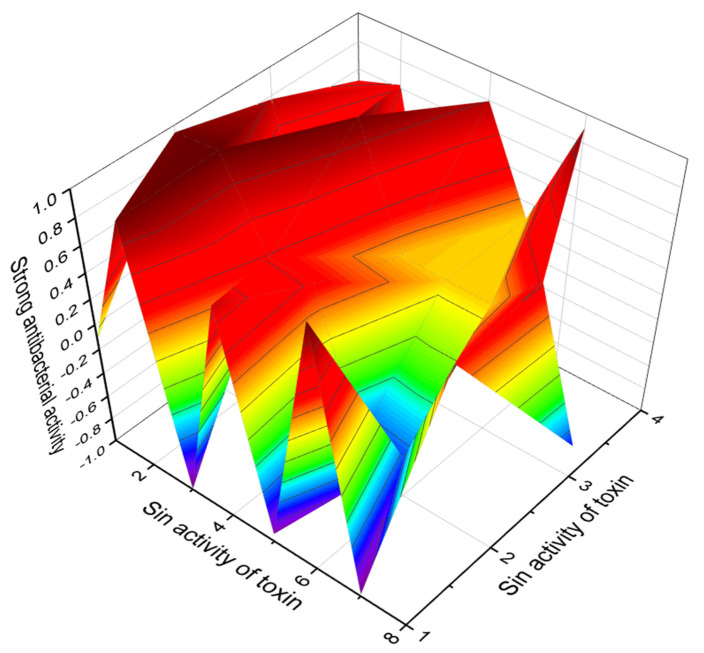
3D graph showing the predicted and calculated strong antibacterial activity of polyether compounds (compound numbers: **108**, **112**, **119**, and **121**) with the highest degree of confidence, more than 99%.

**Figure 39 marinedrugs-20-00292-f039:**
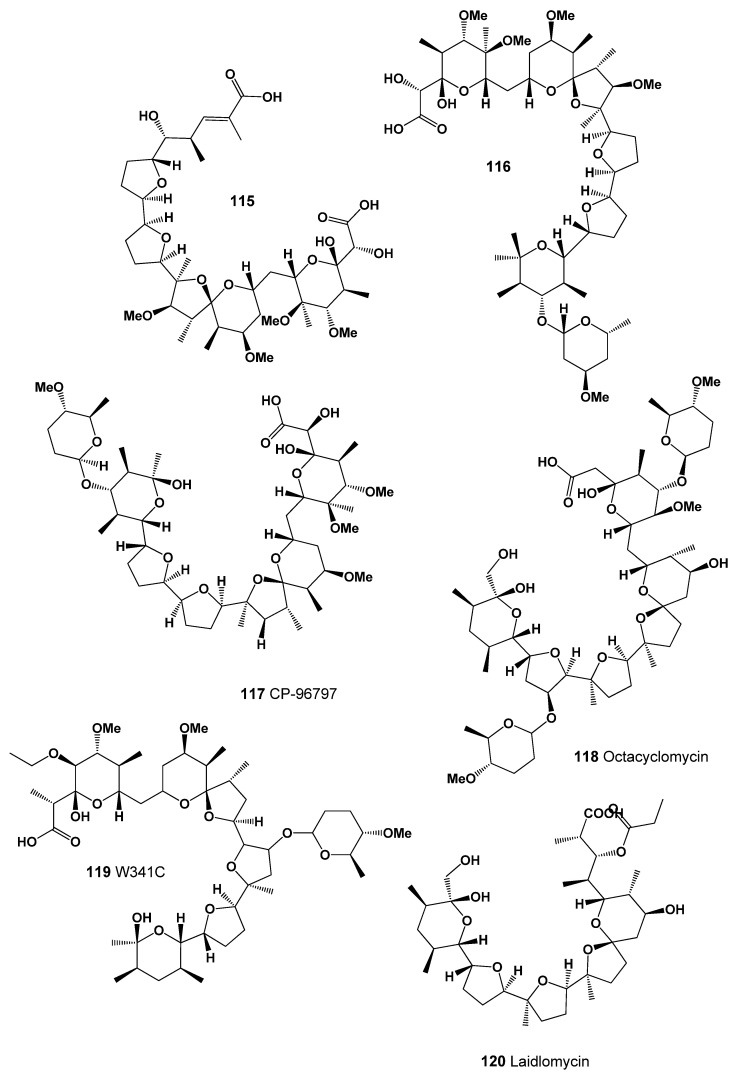
Lipophilic polyether antibiotics produced by Actinomycetes (**115**–**120**).

**Figure 40 marinedrugs-20-00292-f040:**
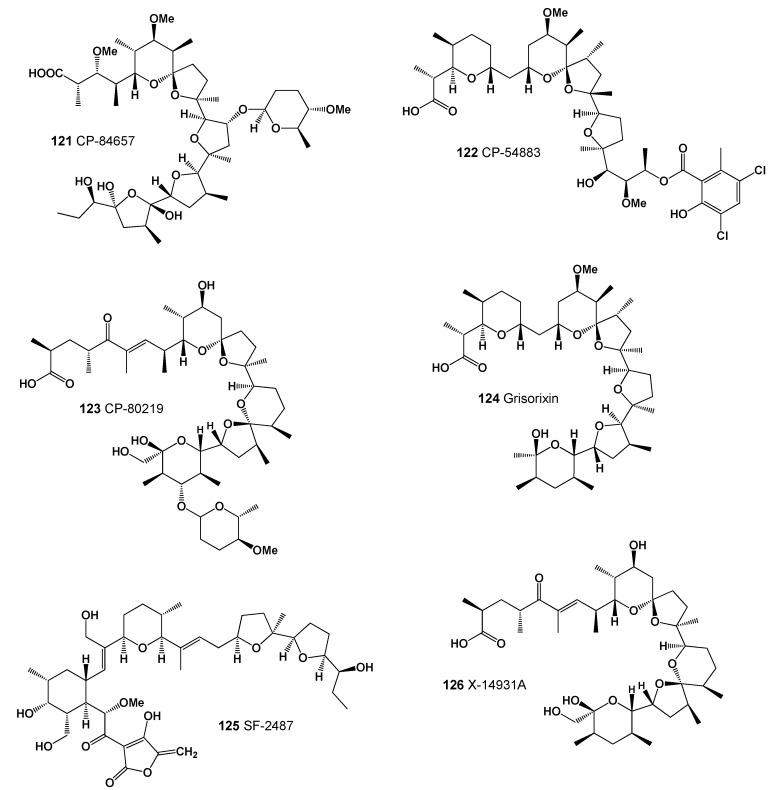
Lipophilic polyether antibiotics produced by Actinomycetes (**121**–**126**).

**Figure 41 marinedrugs-20-00292-f041:**
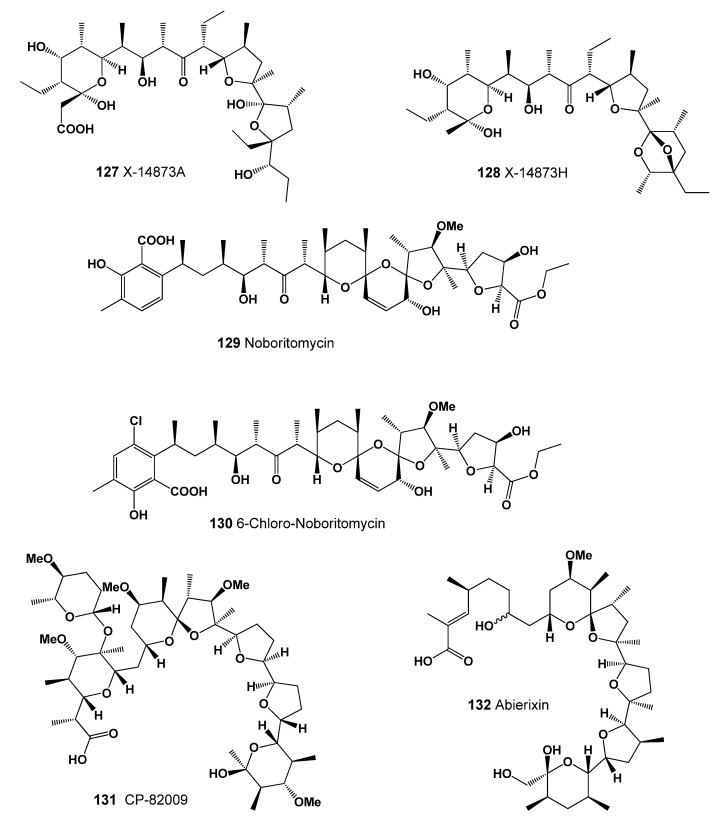
Lipophilic polyether antibiotics produced by Actinomycetes (**127**–**132**).

**Figure 42 marinedrugs-20-00292-f042:**
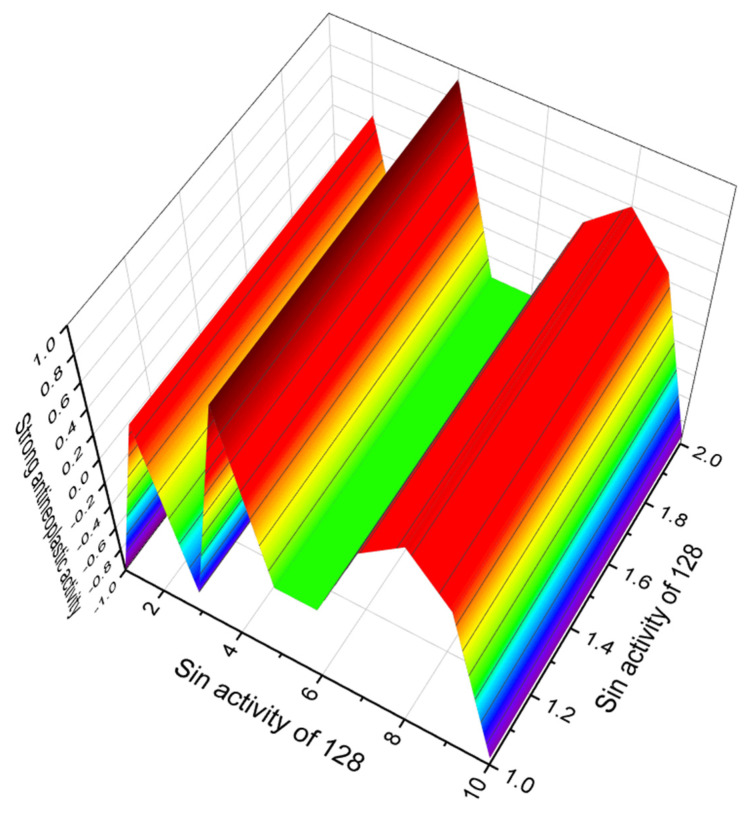
3D graph showing the predicted and calculated strong antineoplastic activity of polyether toxin (compound number: **128**) showing the highest degree of confidence, more than 98%.

**Figure 43 marinedrugs-20-00292-f043:**
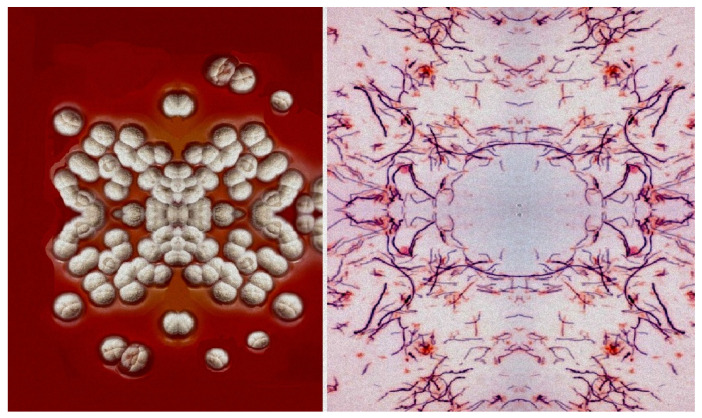
The genus *Actinomadura* is aerobic, Gram-positive, and chemoorganotrophic and belongs to the Thermomonosporaceae family, which includes over 70 species. The *Actinomadura* species produces a wide variety of antibiotics including polyether ionophores. Colonies of *Actinomadura meyerae* are shown in the photographs. Both pictures adapted by author.

**Figure 44 marinedrugs-20-00292-f044:**
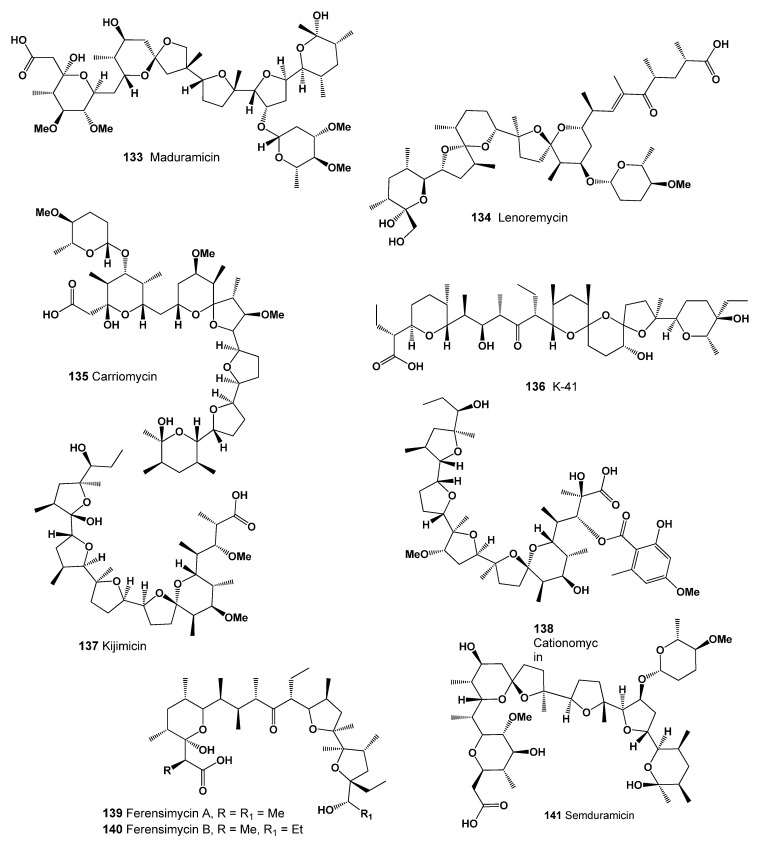
Lipophilic polyether antibiotics produced by Actinomycetes (**133**–**141**).

**Figure 45 marinedrugs-20-00292-f045:**
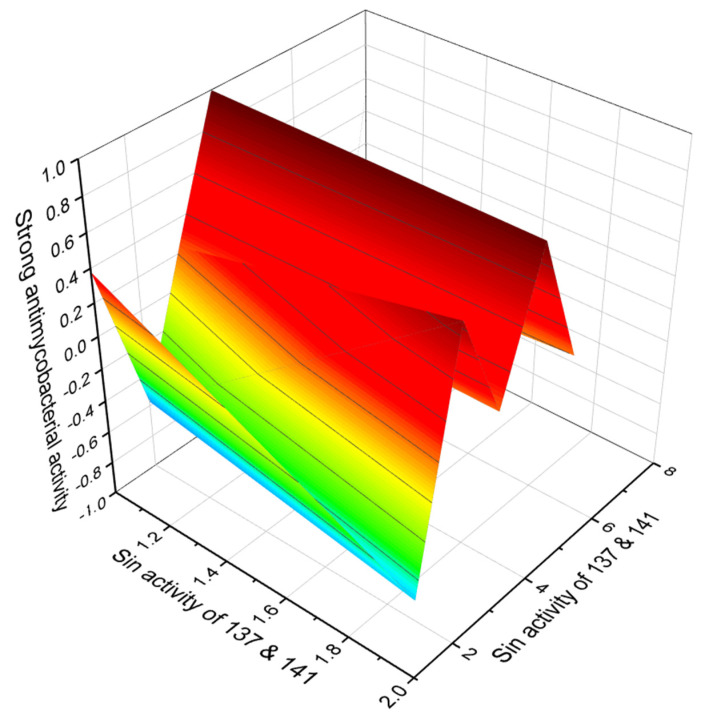
3D graph showing the predicted and calculated strong antimycobacterial activity of polyether compounds (compound numbers: **137** and **141**) with the highest degree of confidence, more than 96%.

**Table 1 marinedrugs-20-00292-t001:** Reported and predicted activity of okadaic acid.

No.	Predicted Biological Activity, Pa *	Reported Activity [30,31,32,33,34,35,36,37,38,39,40,41,42]
**1**	Lipid metabolism regulator (0.999) Angiogenesis stimulant (0.995) DNA synthesis inhibitor (0.991) Apoptosis agonist (0.979) Antineoplastic (0.961) Antifungal (0.844) Antiparasitic (0.843) Antibacterial (0.792) Antineoplastic metabolite (0.763) Cardiotonic (0.719) Immunosuppressant (0.719) Antimitotic (0.694) Growth stimulant (0.650) Platelet aggregation inhibitor (0.539) Hypolipemic (0.538)	Angiogenesis inducer Anticancer Antifungal Antimitotic Antiparasitic Apoptosis inducer Immunosuppressive activity Platelet aggregation inhibitor Protein phosphatase inhibitor

* Only activities with Pa > 0.5 are shown.

**Table 2 marinedrugs-20-00292-t002:** Reported and predicted activity of lipophilic polyether toxins.

No.	Predicted Biological Activity, Pa *	Reported Activity [41,42,43,44,45,46,47,48,49,50,51,52,53,54,55,56,57,58,59,60,61,62,63]
**2**	Lipid metabolism regulator (0.998) Angiogenesis stimulant (0.995) DNA synthesis inhibitor (0.987) Apoptosis agonist (0.984) Antineoplastic (0.964) Antifungal (0.863) Antiparasitic (0.846) Antibacterial (0.818)	Anticancer Apoptosis inducer
**3**	Lipid metabolism regulator (0.998) Angiogenesis stimulant (0.995) Apoptosis agonist (0.983) DNA synthesis inhibitor (0.973) Antineoplastic (0.961)	Anticancer Apoptosis inducer
**4**	Lipid metabolism regulator (0.999) Angiogenesis stimulant (0.994) DNA synthesis inhibitor (0.983) Apoptosis agonist (0.979) Antineoplastic (0.956)	Anticancer
**5**	Lipid metabolism regulator (0.999) Angiogenesis stimulant (0.994) DNA synthesis inhibitor (0.988) Apoptosis agonist (0.974) Antineoplastic (0.959)	Anticancer
**6**	Angiogenesis inhibitor (0.979) Antifungal (0.922) Apoptosis agonist (0.906) Antibacterial (0.900)	Antifungal
**7**	Apoptosis agonist (0.867) Antineoplastic (0.819) Growth stimulant (0.799)	Neurological activity Anticancer
**8**	Antineoplastic (0.846) Apoptosis agonist (0.807) Antifungal (0.796)	Neurological activity Anticancer
**9**	Apoptosis agonist (0.879) Antineoplastic (0.822)	Neurological activity Anticancer

* Only activities with Pa > 0.5 are shown.

**Table 3 marinedrugs-20-00292-t003:** Predicted activity of lipophilic polyether toxins.

No.	Predicted Biological Activity, Pa *
**10**	Antineoplastic (0.945); Apoptosis agonist (0.719); Antineoplastic (sarcoma) (0.664) Antineoplastic (renal cancer) (0.647); T cell inhibitor (0.604); Antineoplastic (pancreatic cancer) (0.594); Antimetastatic (0.590); Antineoplastic (lymphocytic leukemia) (0.581)
**11**	Antineoplastic (0.933); Antimitotic (0.747); Apoptosis agonist (0.648) Antineoplastic (sarcoma) (0.639); Antineoplastic (renal cancer) (0.625) Antineoplastic (pancreatic cancer) (0.579); Antimetastatic (0.570) Antineoplastic (lymphocytic leukemia) (0.559); Antineoplastic (multiple myeloma) (0.534)
**12**	Antineoplastic (0.936); Apoptosis agonist (0.676); Antineoplastic (sarcoma) (0.656) Antineoplastic (renal cancer) (0.636); Antineoplastic (pancreatic cancer) (0.587) Antineoplastic (lymphocytic leukemia) (0.582); Antineoplastic (myeloid leukemia) (0.564)
**13**	Antineoplastic (0.936); Apoptosis agonist (0.717); Antineoplastic (sarcoma) (0.640) Antineoplastic (renal cancer) (0.624); Antineoplastic (pancreatic cancer) (0.579) Antimetastatic (0.575); Antineoplastic (lymphocytic leukemia) (0.551)
**14**	Antineoplastic (0.958); Apoptosis agonist (0.835); Antineoplastic (pancreatic cancer) (0.561)
**15**	Antineoplastic (0.954); Apoptosis agonist (0.835); Antineoplastic (pancreatic cancer) (0.555)
**16**	Antineoplastic (0.917); Antineoplastic (sarcoma) (0.639); Antineoplastic (renal cancer) (0.637) Antineoplastic (lymphocytic leukemia) (0.545); Apoptosis agonist (0.544)
**17**	Antineoplastic (0.919); Apoptosis agonist (0.765); Antimitotic (0.739)
**18**	Antineoplastic (0.928); Apoptosis agonist (0.748); Antineoplastic (renal cancer) (0.715) Antineoplastic (lymphocytic leukemia) (0.654); Prostate cancer treatment (0.500)

* Only activities with Pa > 0.5 are shown.

**Table 4 marinedrugs-20-00292-t004:** Predicted activity of lipophilic polyether toxins (**19**–**34**).

No.	Predicted Biological Activity, Pa *
**19**	Antineoplastic (0.899); Apoptosis agonist (0.849); Antimetastatic (0.593)
**20**	Antineoplastic (0.901); Apoptosis agonist (0.845); Antimetastatic (0.602)
**21**	Antineoplastic (0.940); Apoptosis agonist (0.681); Antineoplastic (renal cancer) (0.664)
**22**	Antineoplastic (0.933); Antifungal (0.925); Antibacterial (0.904); Antiparasitic (0.874) Apoptosis agonist (0.679); Antineoplastic (renal cancer) (0.653); Antimetastatic (0.621)
**23**	Antifungal (0.933); Antineoplastic (0.933); Antibacterial (0.903); Antiparasitic (0.863) Antimitotic (0.807); Antibiotic (0.791); Apoptosis agonist (0.654) Antineoplastic (renal cancer) (0.650); Antimetastatic (0.633)
**24**	Antifungal (0.929); Antineoplastic (0.926); Antibacterial (0.914); Apoptosis agonist (0.652) Antineoplastic (renal cancer) (0.640); Antimetastatic (0.623); Antileukemic (0.591)
**25**	Antineoplastic (0.941); Apoptosis agonist (0.773); Antineoplastic (renal cancer) (0.665)
**26**	Antineoplastic (0.947); Apoptosis agonist (0.815); Antineoplastic (renal cancer) (0.693)
**27**	Antineoplastic (0.951); Apoptosis agonist (0.757); Antineoplastic (renal cancer) (0.732) Alzheimer’s disease treatment (0.631); Antileukemic (0.626); Antimetastatic (0.607)
**28**	Antibacterial (0.979); Antifungal (0.965); Antineoplastic (0.931); Antimetastatic (0.565)
**29**	Antineoplastic (0.921); Apoptosis agonist (0.664); Antimetastatic (0.651)
**30**	Antineoplastic (0.929); Cytostatic (0.922); Apoptosis agonist (0.678) Antineoplastic (lymphocytic leukemia) (0.672); Antineoplastic (renal cancer) (0.606)
**31**	Antineoplastic (0.931); Cytostatic (0.899); Antifungal (0.897); Antiparasitic (0.882) Apoptosis agonist (0.664); Antimetastatic (0.655); Antineoplastic (renal cancer) (0.612)
**32**	Antineoplastic (0.918); Antifungal (0.884); Cytostatic (0.859); Antibacterial (0.849) Antimitotic (0.817); Antiparasitic (0.793); Antineoplastic (renal cancer) (0.689) Antimetastatic (0.653); Apoptosis agonist (0.616); Prostate cancer treatment (0.500)
**33**	Antineoplastic (0.926); Cytostatic (0.907); Antifungal (0.878); Antibacterial (0.850) Antimitotic (0.821); Antineoplastic (renal cancer) (0.698); Antimetastatic (0.654) Antileukemic (0.653); Apoptosis agonist (0.632); Prostate cancer treatment (0.526)
**34**	Antineoplastic (0.926); Antifungal (0.885); Cytostatic (0.876); Antibacterial (0.829) Antiparasitic (0.820); Antimitotic (0.815); Antineoplastic (renal cancer) (0.700) Antileukemic (0.684); Antimetastatic (0.655); Apoptosis agonist (0.611)

* Only activities with Pa > 0.5 are shown.

**Table 5 marinedrugs-20-00292-t005:** Predicted activity of lipophilic polyether toxins (**35**–**45**).

No.	Predicted Biological Activity, Pa *
**35**	Antineoplastic (0.953); Antibacterial (0.907); Antifungal (0.814); Antimitotic (0.721) Apoptosis agonist (0.675); Antiparasitic (0.651); Antimetastatic (0.628)
**36**	Antineoplastic (0.936); Antibacterial (0.930); Antifungal (0.848); Apoptosis agonist (0.623) Antimetastatic (0.609); Anti-mycoplasmal (0.577)
**37**	Cytostatic (0.912); Antineoplastic (0.912); Antifungal (0.901); Antibacterial (0.860) Antiparasitic (0.849); Antimitotic (0.825); Antineoplastic (renal cancer) (0.656) Antimetastatic (0.655); Apoptosis agonist (0.642)
**38**	Cytostatic (0.961); Antineoplastic (0.925); Antifungal (0.910); Antibacterial (0.871) Antiparasitic (0.868); Antimitotic (0.846); Apoptosis agonist (0.694); Antimetastatic (0.672)
**39**	Antineoplastic (0.913); Cytostatic (0.910); Antifungal (0.870); Antiparasitic (0.852) Antibacterial (0.842); Antimitotic (0.822); Apoptosis agonist (0.683); Antimetastatic (0.654) Antineoplastic (renal cancer) (0.652); Autoimmune disorders treatment (0.565)
**40**	Antineoplastic (0.949); Antineoplastic (liver cancer) (0.905); Antiparasitic (0.869) Cytostatic (0.859); Apoptosis agonist (0.840); Antimetastatic (0.739)
**41**	Antineoplastic (0.946); Antimitotic (0.832); Antifungal (0.788); Antiparasitic (0.785) Antineoplastic (renal cancer) (0.750); Antimetastatic (0.691); Prostate cancer treatment (0.521)
**42**	Antineoplastic (0.920); Antifungal (0.821); Antibacterial (0.805); Antimetastatic (0.526)
**43**	Antineoplastic (0.928); Apoptosis agonist (0.824); Antimetastatic (0.515)
**44**	Antineoplastic (0.930); Antimitotic (0.823); Apoptosis agonist (0.762) Alzheimer’s disease treatment (0.741); Antineoplastic (renal cancer) (0.520)
**45**	Antineoplastic (0.967); Growth stimulant (0.881); Antiparasitic (0.845); Antibacterial (0.832) Antimitotic (0.821); Antifungal (0.775); Apoptosis agonist (0.699); Cytostatic (0.687) Antimetastatic (0.655); Antineoplastic (non-Hodgkin’s lymphoma) (0.587)

* Only activities with Pa > 0.5 are shown.

**Table 6 marinedrugs-20-00292-t006:** Predicted activity of lipophilic polyether toxins (**46**–**61**).

No.	Predicted Biological Activity, Pa *
**46**	Antineoplastic (0.871); Apoptosis agonist (0.656); Antibacterial (0.631)
**47**	Antineoplastic (0.839); Antibacterial (0.720); Antineoplastic (myeloid leukemia) (0.551) Antileukemic (0.540); Antineoplastic (lymphocytic leukemia) (0.531)
**48**	Antineoplastic (0.855); Antifungal (0.793); Antibacterial (0.704); Antimitotic (0.674) Antineoplastic (myeloid leukemia) (0.562); Antineoplastic (lymphocytic leukemia) (0.553)
**49**	Antineoplastic (0.858); Antifungal (0.798); Antibacterial (0.743); Apoptosis agonist (0.631) Antineoplastic (myeloid leukemia) (0.573); Antineoplastic (lymphocytic leukemia) (0.550)
**50**	Antineoplastic (0.906); Antifungal (0.813); Antibacterial (0.748) Apoptosis agonist (0.692); Alzheimer’s disease treatment (0.625) Antineoplastic (myeloid leukemia) (0.576); Antineoplastic (lymphocytic leukemia) (0.560)
**51**	Antineoplastic (0.871); Antifungal (0.736); Antimitotic (0.685); Antibacterial (0.683) Apoptosis agonist (0.670); Antineoplastic (myeloid leukemia) (0.606) Antileukemic (0.593); Antineoplastic (lymphocytic leukemia) (0.568)
**52**	Antineoplastic (0.886); Antifungal (0.768); Antimitotic (0.731); Antibacterial (0.716) Apoptosis agonist (0.644); Antineoplastic (myeloid leukemia) (0.581) Antileukemic (0.580); Antineoplastic (lymphocytic leukemia) (0.544)
**53**	Antineoplastic (0.851); Antifungal (0.804); Antibacterial (0.735); Antimitotic (0.704) Apoptosis agonist (638); Antineoplastic (myeloid leukemia) (0.566) Antileukemic (0.554); Antineoplastic (lymphocytic leukemia) (0.548)
**54**	Antineoplastic (0.928); Antibacterial (0.854); Antifungal (0.827); Antimitotic (0.655) Antimetastatic (0.615); Prostate cancer treatment (0.613); Apoptosis agonist (0.591)
**55**	Antineoplastic (0.914); Antifungal (0.820); Antimitotic (0.653); Antimetastatic (0.620) Prostate cancer treatment (0.612); Antineoplastic (lymphocytic leukemia) (0.541)
**56**	Antineoplastic (0.915); Antifungal (0.752); Antibacterial (0.746); Antimitotic (0.698) Antineoplastic (myeloid leukemia) (0.677); Antimetastatic (0.613) Prostate cancer treatment (0.592); Antineoplastic (lymphocytic leukemia) (0.550)
**57**	Antineoplastic (0.872); Antibacterial (0.793); Antifungal (0.668); Antimitotic (0.662) Antimetastatic (0.656); Antineoplastic (lymphocytic leukemia) (0.576) Apoptosis agonist (0.509); Antineoplastic (non-Hodgkin’s lymphoma) (0.506)
**58**	Antineoplastic (0.921); Antibacterial (0.877); Antifungal (0.852); Antimitotic (0.704) Prostate cancer treatment (0.625); Antimetastatic (0.621) Antineoplastic (lymphocytic leukemia) (0.575); Antineoplastic (myeloid leukemia) (0.533)
**59**	Antineoplastic (0.895); Antibacterial (0.856); Antifungal (0.796); Antimitotic (0.762) Antineoplastic (lymphocytic leukemia) (0.591); Antineoplastic (myeloid leukemia) (0.545)
**60**	Antineoplastic (0.935); Antibacterial (0.912); Antifungal (0.905); Chemopreventive (0.760) Apoptosis agonist (0.638); Anticarcinogenic (0.625); Antimetastatic (0.617)
**61**	Antibacterial (0.934); Antineoplastic (0.933); Antifungal (0.913); Chemopreventive (0.758)

* Only activities with Pa > 0.5 are shown.

**Table 7 marinedrugs-20-00292-t007:** Predicted activity of lipophilic polyether toxins (**62**–**72**).

No.	Predicted Biological Activity, Pa *
**62**	Antifungal (0.884); Lipid metabolism regulator (0.875); Apoptosis agonist (0.851)
**63**	Antifungal (0.890); Antibacterial (0.837); Antineoplastic (0.836); Apoptosis agonist (0.782)
**64**	Antifungal (0.842); Antineoplastic (0.814); Antibacterial (0.762); Apoptosis agonist (0.738)
**65**	Antifungal (0.896); Antineoplastic (0.867); Antibacterial (0.831); Apoptosis agonist (0.806)
**66**	Antifungal (0.901); Antibacterial (0.870); Antineoplastic (0.863); Apoptosis agonist (0.718)
**67**	Antifungal (0.895); Apoptosis agonist (0.858); Antineoplastic (0.843); Antibacterial (0.826)
**68**	Antifungal (0.892); Antineoplastic (0.794); Antibacterial (0.773); Apoptosis agonist (0.772)
**69**	Antifungal (0.904); Antineoplastic (0.850); Apoptosis agonist (0.849); Antibacterial (0.821)
**70**	Antifungal (0.896); Antineoplastic (0.800); Apoptosis agonist (0.781); Antibacterial (0.779)
**71**	Antifungal (0.907); Antineoplastic (0.795); Antibacterial (0.783); Apoptosis agonist (0.777)
**72**	Antifungal (0.901); Antineoplastic (0.789); Antibacterial (0.780); Apoptosis agonist (0.772)

* Only activities with Pa > 0.5 are shown.

**Table 8 marinedrugs-20-00292-t008:** Predicted activity of lipophilic polyether toxins (**73**–**76**).

No.	Predicted Biological Activity, Pa *
**73**	Antifungal (0.974); Antibacterial (0.969); Antineoplastic (0.906); Antimetastatic (0.524)
**74**	Antifungal (0.979); Antibacterial (0.959); Antineoplastic (0.916); Apoptosis agonist (0.672)
**75**	Antifungal (0.973); Antibacterial (0.939); Antineoplastic (0.870); Apoptosis agonist (0.628)
**76**	Antifungal (0.975); Antibacterial (0.943); Antineoplastic (0.879); Apoptosis agonist (0.638)

* Only activities with Pa > 0.5 are shown.

**Table 9 marinedrugs-20-00292-t009:** Predicted activity of lipophilic polyether toxins (**75**–**85**).

No.	Predicted Biological Activity, Pa *
**75**	Antifungal (0.973); Antibacterial (0.939); Antineoplastic (0.870); Apoptosis agonist (0.628)
**76**	Antifungal (0.975); Antibacterial (0.943); Antineoplastic (0.879); Apoptosis agonist (0.638)
**77**	Antiprotozoal (Coccidial) (0.943); Antibacterial (0.939); Antiparasitic (0.867)
**78**	Antiprotozoal (Coccidial) (0.928); Antibacterial (0.921); Antiparasitic (0.872)
**79**	Antiprotozoal (Coccidial) (0.906); Antibacterial (0.902); Antiparasitic (0.877)
**80**	Antibacterial (0.953); Antiprotozoal (Coccidial) (0.924); Antitreponemal (0.906) Antiparasitic (0.898); Antineoplastic (0.867); Antiprotozoal (Plasmodium) (0.686)
**81**	Antineoplastic (0.893); Antifungal (0.842); Antibacterial (0.818) Apoptosis agonist (0.638); Antiprotozoal (Coccidial) (0.634); Antimetastatic (0.629)
**82**	Growth stimulant (0.974); Antimycobacterial (0.971); Antifungal (0.967) Antiprotozoal (Coccidial) (0.943); Antibacterial (0.942); Antineoplastic (0.937) Antiparasitic (0.919); Antiprotozoal (Plasmodium) (0.699); Antimetastatic (0.602)
**83**	Growth stimulant (0.974); Antimycobacterial (0.972); Antifungal (0.971) Antiprotozoal (Coccidial) (0.947); Antibacterial (0.943); Antineoplastic (0.937) Antiparasitic (0.921); Antiprotozoal (Plasmodium) (0.701); Antimetastatic (0.604) Antineoplastic (renal cancer) (0.604); Antineoplastic (sarcoma) (0.590) Apoptosis agonist (0.580); Antineoplastic (lymphocytic leukemia) (0.559)
**84**	Growth stimulant (0.974); Antimycobacterial (0.972); Antifungal (0.971) Antiprotozoal (Coccidial) (0.947); Antibacterial (0.943); Antineoplastic (0.937) Antiparasitic (0.921); Antiprotozoal (Plasmodium) (0.701); Antimitotic (0.663)
**85**	Antimycobacterial (0.963); Growth stimulant (0.960); Antineoplastic (0.956) Antifungal (0.951); Antibacterial (0.942); Antiprotozoal (Coccidial) (0.937) Antiparasitic (0.917); Apoptosis agonist (0.914); Antiprotozoal (Plasmodium) (0.701)

* Only activities with Pa > 0.5 are shown.

**Table 10 marinedrugs-20-00292-t010:** Predicted activity of lipophilic polyether toxins (**86**–**97**).

No.	Predicted Biological Activity, Pa *
**86**	Growth stimulant (0.932); Antitreponemal (0.908); Antineoplastic (0.906) Antiprotozoal (Coccidial) (0.900); Antiparasitic (0.891); Antifungal (0.882)
**87**	Growth stimulant (0.932); Antitreponemal (0.910); Antineoplastic (0.906) Antiprotozoal (Coccidial) (0.900); Antiparasitic (0.891); Antifungal (0.877)
**88**	Antieczematic (0.920); Antibacterial (0.839); Antineoplastic (0.802)
**89**	Antieczematic (0.922); Antibacterial (0.834); Antineoplastic (0.820)
**90**	Antineoplastic (0.929); Growth stimulant (0.920); Antibacterial (0.918) Antiprotozoal (Coccidial) (0.903); Antifungal (0.898); Antiparasitic (0.877) Antimycobacterial (0.790); Antineoplastic (renal cancer) (0.626); Antimetastatic (0.624)
**91**	Antineoplastic (0.928); Growth stimulant (0.919); Antibacterial (0.919) Antiprotozoal (Coccidial) (0.903); Antifungal (0.895); Antiparasitic (0.878) Antimycobacterial (0.660); Antimitotic (0.659); Antineoplastic (renal cancer) (0.623) Antimetastatic (0.622); Antiprotozoal (Plasmodium) (0.547); Chemoprotective (0.523)
**92**	Antibacterial (0.896); Growth stimulant (0.886); Antiprotozoal (Coccidial) (0.877) Antineoplastic (0.868); Antifungal (0.854); Antiparasitic (0.747)
**93**	Antibacterial (0.945); Antiprotozoal (Coccidial) (0.938); Growth stimulant (0.929) Antiparasitic (0.916); Antitreponemal (0.915); Antiprotozoal (Plasmodium) (0.731)
**94**	Antibacterial (0.944); Antiprotozoal (Coccidial) (0.940); Growth stimulant (0.918) Antitreponemal (0.917); Antiparasitic (0.915); Antineoplastic (0.900); Antifungal (0.858) Antiprotozoal (Plasmodium) (0.732); Antineoplastic (renal cancer) (0.677) Antimycoplasmal (0.673); Antineoplastic (lymphocytic leukemia) (0.583)
**95**	Antibacterial (0.912); Antineoplastic (0.873); Antiprotozoal (Coccidial) (0.865) Antiparasitic (0.796); Antifungal (0.794); Antineoplastic (renal cancer) (0.633) Antineoplastic (sarcoma) (0.621); Antimetastatic (0.578) Antineoplastic (lymphocytic leukemia) (0.551)
**96**	Antibacterial (0.992); Antiprotozoal (Coccidial) (0.964); Antitreponemal (0.954) Antiparasitic (0.948); Antineoplastic (0.901); Antifungal (0.796) Antiprotozoal (Plasmodium) (0.790); Antineoplastic (renal cancer) (0.575)
**97**	Antibacterial (0.981); Antiprotozoal (Coccidial) (0.978); Antitreponemal (0.966) Antiparasitic (0.946); Antineoplastic (0.919); Growth stimulant (0.842) Antiprotozoal (Plasmodium) (0.696); Antimycoplasmal (0.597)

* Only activities with Pa > 0.5 are shown.

**Table 11 marinedrugs-20-00292-t011:** Predicted activity of lipophilic polyether toxins (**98**–**107**).

No.	Predicted Biological Activity, Pa *
**98**	Antibacterial (0.992); Antiprotozoal (Coccidial) (0.977); Antitreponemal (0.959) Antiparasitic (0.950); Antifungal (0.799); Antiprotozoal (Plasmodium) (0.778)
**99**	Antibacterial (0.991); Antiprotozoal (Coccidial) (0.964); Antitreponemal (0.953) Antiparasitic (0.950); Antiprotozoal (Plasmodium) (0.803); Antifungal (0.798)
**100**	Antimycobacterial (0.957); Antineoplastic (0.956); Antifungal (0.913) Antibacterial (0.910); Antiparasitic (0.876); Antihelmintic (0.804)
**101**	Growth stimulant (0.946); Antibacterial (0.921); Antiprotozoal (Coccidial) (0.918) Antiparasitic (0.909); Antineoplastic (0.905); Antifungal (0.881) Antiprotozoal (0.880); Antitreponemal (0.829); Antiprotozoal (Plasmodium) (0.702)
**102**	Antiprotozoal (0.944); Antibacterial (0.935); Antiprotozoal (Coccidial) (0.931) Antiparasitic (0.927); Antitreponemal (0.914); Antifungal (0.879) Antiprotozoal (Plasmodium) (0.725); Antimycoplasmal (0.603)
**103**	Antiprotozoal (0.948); Antiprotozoal (Coccidial) (0.938); Antibacterial (0.935) Antitreponemal (0.930); Antiparasitic (0.927); Antifungal (0.880); Antihelmintic (0.704) Antiprotozoal (Plasmodium) (0.690); Antimycoplasmal (0.650)
**104**	Antibacterial (0.946); Antiprotozoal (Coccidial) (0.943); Antiparasitic (0.932) Antitreponemal (0.924); Antifungal (0.890); Antiprotozoal (Plasmodium) (0.692)
**105**	Antibacterial (0.930); Antitreponemal (0.926); Antiprotozoal (Coccidial) (0.922) Antiparasitic (0.914); Antifungal (0.858); Antiprotozoal (Plasmodium) (0.637)
**106**	Antineoplastic (0.919); Antifungal (0.894); Antiparasitic (0.872); Antibacterial (0.869)
**107**	Antibacterial (0.995); Antiprotozoal (Coccidial) (0.988); Antineoplastic (0.926) Antiparasitic (0.925); Growth stimulant (0.904); Antifungal (0.898) Antitreponemal (0.803); Antimitotic (0.706); Antihelmintic (0.702)

* Only activities with Pa > 0.5 are shown.

**Table 12 marinedrugs-20-00292-t012:** Predicted activity of lipophilic polyether toxins (**108**–**121**).

No.	Predicted Biological Activity, Pa *
**108**	Antibacterial (0.996); Antiparasitic (0.994); Antiprotozoal (Coccidial) (0.985) Antineoplastic (0.937); Antifungal (0.935); Antitreponemal (0.919) Antiprotozoal (Plasmodium) (0.769); Antimycoplasmal (0.570)
**109**	Antibacterial (0.936); Antiprotozoal (Coccidial) (0.890); Antiparasitic (0.791) Antifungal (0.789); Antiprotozoal (Plasmodium) (0.507); Antimycoplasmal (0.500)
**110**	Antibacterial (0.991); Antiprotozoal (Coccidial) (0.990); Antiparasitic (0.924) Antitreponemal (0.921); Antineoplastic (0.915); Antiprotozoal (Plasmodium) (0.746)
**111**	Antimycobacterial (0.969); Antibacterial (0.936); Antifungal (0.907) Antiprotozoal (Coccidial) (0.897); Antiparasitic (0.866); Antimycoplasmal (0.533)
**112**	Antibacterial (0.994); Antiprotozoal (Coccidial) (0.969); Antiparasitic (0.968) Antineoplastic (0.932); Antifungal (0.924); Antitreponemal (0.904) Antiprotozoal (Plasmodium) (0.703); Antimycoplasmal (0.562)
**113**	Antibacterial (0.946); Antiprotozoal (Coccidial) (0.943); Antiparasitic (0.921) Antitreponemal (0.915); Antifungal (0.875); Antiprotozoal (Plasmodium) (0.693)
**114**	Growth stimulant (0.932); Antitreponemal (0.908); Antineoplastic (0.906) Antiprotozoal (Coccidial) (0.900); Antiparasitic (0.891); Antifungal (0.882)
**115**	Antiprotozoal (Coccidial) (0.916); Antibacterial (0.912); Antiparasitic (0.907)
**116**	Antibacterial (0.955); Antiprotozoal (Coccidial) (0.929); Antitreponemal (0.916)
**117**	Antibacterial (0.982); Antiprotozoal (Coccidial) (0.977); Antitreponemal (0.952) Antiparasitic (0.941); Antifungal (0.831); Antiprotozoal (Plasmodium) (0.786)
**118**	Antibacterial (0.995); Antiprotozoal (Coccidial) (0.986); Antineoplastic (0.936) Antiparasitic (0.921); Growth stimulant (0.907); Antifungal (0.884) Antitreponemal (0.843); Antiprotozoal (Plasmodium) (0.708); Antimycoplasmal (0.647)
**119**	Antibacterial (0.995); Antiprotozoal (Coccidial) (0.984); Antiparasitic (0.950) Antineoplastic (0.930); Growth stimulant (0.889); Antitreponemal (0.886) Antifungal (0.842); Antiprotozoal (Plasmodium) (0.723); Antimycoplasmal (0.582)
**120**	Growth stimulant (0.945); Antiprotozoal (Coccidial) (0.922); Antibacterial (0.921) Antiparasitic (0.916); Antitreponemal (0.775); Antiprotozoal (Plasmodium) (0.729)
**121**	Antibacterial (0.987); Antiprotozoal (Coccidial) (0.962); Antineoplastic (0.947) Antifungal (0.907); Antiparasitic (0.889); Antiprotozoal (Plasmodium) (0.686)

* Only activities with Pa > 0.5 are shown.

**Table 13 marinedrugs-20-00292-t013:** Predicted activity of lipophilic polyether toxins (**122**–1**32**).

No.	Predicted Biological Activity, Pa *
**122**	Antibacterial (0.944); Antiprotozoal (Coccidial) (0.940); Antitreponemal (0.917) Antiparasitic (0.915); Antifungal (0.858); Antiprotozoal (Plasmodium) (0.732)
**123**	Antiprotozoal (Coccidial) (0.894); Antiparasitic (0.893); Antibacterial (0.876)
**124**	Antineoplastic (0.907); Antibacterial (0.893); Antifungal (0.863); Apoptosis agonist (0.799)
**125**	Antibacterial (0.944); Antiparasitic (0.941); Antiprotozoal (Coccidial) (0.935) Antitreponemal (0.925); Antifungal (0.905); Antiprotozoal (Plasmodium) (0.764)
**126**	Antiparasitic (0.995); Antibacterial (0.995); Antiprotozoal (Coccidial) (0.986) Antifungal (0.935); Antitreponemal (0.933); Antiprotozoal (Plasmodium) (0.776)
**127**	Antibacterial (0.925); Antineoplastic (0.907); Antiprotozoal (Coccidial) (0.899)
**128**	Antineoplastic (0.985); Chemoprotective (0.964); Growth stimulant (0.934) Antitreponemal (0.913); Antiparasitic (0.902); Antifungal (0.902); Antiprotozoal (Coccidial) (0.895) Antibacterial (0.874); Antihelmintic (0.660); Antiprotozoal (Plasmodium) (0.602)
**129**	Antineoplastic (0.932); Growth stimulant (0.926); Antibacterial (0.912) Antiprotozoal (Coccidial) (0.910); Antifungal (0.897); Antiparasitic (0.878) Antimycobacterial (0.775); Antitreponemal (0.676); Antimetastatic (0.629)
**130**	Antineoplastic (0.924); Growth stimulant (0.922); Antiprotozoal (Coccidial) (0.910) Antibacterial (0.904); Antifungal (0.898); Antiparasitic (0.880); Antimycobacterial (0.800)
**131**	Antibacterial (0.981); Antiprotozoal (Coccidial) (0.978); Antitreponemal (0.961) Antiparasitic (0.950); Antineoplastic (0.930); Antiprotozoal (Plasmodium) (0.646)
**132**	Antibacterial (0.949); Antiprotozoal (Coccidial) (0.911); Growth stimulant (0.909) Antiparasitic (0.899); Antifungal (0.894); Antitreponemal (0.841)

* Only activities with Pa > 0.5 are shown.

**Table 14 marinedrugs-20-00292-t014:** Predicted activity of lipophilic polyether toxins (**133**–**141**).

No.	Predicted Biological Activity, Pa *
**133**	Antibacterial (0.975); Antineoplastic (0.948); Antiprotozoal (Coccidial) (0.935) Antiparasitic (0.909); Antifungal (0.904); Antihelmintic (0.821); Antitreponemal (0.724) Antimycoplasmal (0.633); Antiprotozoal (Plasmodium) (0.628)
**134**	Antibacterial (0.994); Antiparasitic (0.979); Antiprotozoal (Coccidial) (0.973) Antineoplastic (0.935); Antifungal (0.922); Antitreponemal (0.911) Antiprotozoal (Plasmodium) (0.738); Antimycoplasmal (0.525)
**135**	Antibacterial (0.969); Antiprotozoal (Coccidial) (0.933); Antineoplastic (0.917) Antiparasitic (0.904); Antitreponemal (0.897); Antiprotozoal (Plasmodium) (0.716)
**136**	Growth stimulant (0.965); Antimycobacterial (0.965); Antifungal (0.945) Antineoplastic (0.940); Antiprotozoal (Coccidial) (0.935); Antibacterial (0.935) Antiparasitic (0.921); Antitreponemal (0.861); Antihelmintic (0.855)
**137**	Growth stimulant (0.968); Antimycobacterial (0.967); Antifungal (0.949) Antineoplastic (0.944); Antiprotozoal (Coccidial) (0.933); Antibacterial (0.930) Antiparasitic (0.922); Antiprotozoal (Plasmodium) (0.788)
**138**	Antibacterial (0.940); Antiprotozoal (Coccidial) (0.935); Antineoplastic (0.891) Antifungal (0.880); Antiparasitic (0.848); Antimycoplasmal (0.579)
**139**	Growth stimulant (0.883); Antitreponemal (0.837); Antiprotozoal (Coccidial) (0.829) Antibacterial (0.823); Antiparasitic (0.814); Antiprotozoal (Plasmodium) (0.546)
**140**	Growth stimulant (0.883); Antitreponemal (0.837); Antiprotozoal (Coccidial) (0.829) Antibacterial (0.823); Antiparasitic (0.814); Antiprotozoal (Plasmodium) (0.546)
**141**	Antimycobacterial (0.965); Antibacterial (0.932); Antifungal (0.905) Antiprotozoal (Coccidial) (0.894); Antiparasitic (0.861)

* Only activities with Pa > 0.5 are shown.

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
