# Peer review of "Natural Polyether Ionophores and Their Pharmacological Profile"

_marinedrugs, 2022, doi:10.3390/md20050292_

Round 1
Reviewer 1 Report
The topic of the manuscript under review seems to be promising but the presentation is poor. Following issues may be addressed for the improvement.
- Title of the manuscript is somewhat misleading according to the content of the manuscript and hence, it should be modified.
- if it is a review article then what does following sentence in abstract mean? "QSAR (Quantitative Structure-Activity Relationships) method was used to calculate the pharmacological profile of polyether ionophore antibiotics. According to the data obtained, ........."
- If QSAR was performed in this study then surely its not a kind of review article. All details of methods adopted for QSAR, validation of the used parameters etc must be provided.
- Similarly, Tables represent reported and predicted activity. So, which method was used for the prediction and how it was implemented? All these details should be provided in the experimental section.
- Figures also show predicted and calculated biological property. Authors must clarify.
Author Response
Reviewer 1
Thanks to Reviewer 1 for reading the manuscript and some suggestions he made.
I fundamentally disagree with the reviewer that this is not a review. This is a typical review article that describes the biological activity of natural polyether ionophores. In addition to the already known activities, an additional activity has been added, which was obtained using PASS.
To accommodate the reviewer's request, a short paragraph has been inserted giving all the information about QSAR. We hope that this information will be enough to understand the presented review, and its figures and tables.
Yes, I agree that the title of the review is not very good, it has been changed.
All changes and additions are made right in the text and are highlighted in blue.
Valery Dembitsky
Reviewer 2 Report
The present manuscript shows a compilation of several polyether ionophore natural products. Biological activities are commented for each of them, and in silico prediction of pharmacological activity is additionally presented. In my opinión, the work presented is quite interesting and complete and well deserves publication in Marine drugs. However, I should suggest a few things to be consider by the autor before manuscript is fully accepted.
- I would suggest that the first senteces (sentences 619-626) in the Conclusion section should be stated earlier in the manuscript, and precisely in the introduction section. In my opinión, it would be claryfing to the reader to state from the begining in the manuscript that preditions shown along the manuscript are made with program PASS.
- Table 1 and Table 2 in the manuscript show predicted and reported activity of compounds, while the rest of the tables in the manuscript only show predicted activities. I should suggest including a column in tables 3-14 with data regarding reported activity in the same way as it is done in table 1 and 2. Also interesting would be including a column in such tables with reference numbers wherein reported activity can be found.
Some minor corrections:
- In the abstract, the word “polyester” is used several times. Please correct and change to polyether.
- Page 1 of 61, line 36: “can be synthesized from… “ should be read “can be biosynthesized from… “
- Page 3 of 61, line 67: please include references when stating “ according to published experimental data”. References are not given in the paragraph and neither in Table 1.
- Indicate what “PASS” stands for the first time it appears in the manuscript.
- Page 3 of 61, line 74: okadaic derivatives are shown in Table 2 (not Table 1), please correct.
- Page 3 of 61, Figure 2. IT is stated that all photos are taken from sites where permisison in granted for non-commercial use. I belive that the author should indicate the site from where such photo was taken.
- What is the meaning of the word “sin” in the scales in the 3D graphs??
- I would like to read what is the criteria that the author has followed to choose a concrete activity for the representation of the 3D graphs.
- Include references in Table 1 and Table 2 for the reported activities.
- Page 6 of 61, the word “trans” should be italic.
- Page 6 of 61, line 127: please include reference at the end of the paragraph
- Page 7 of 61, line 132: please indicate Figure number where the ready could see structure of 19. Also consider, including compounds 18 and 19 in the same Figure since they are analogues.
- Page 8 of 61, line 139: is the legend of Figure 6 correct? Prorocenting and gambierol are also pectenotoxins?
- Page 9 of 61, line 150: dinoagellate should be read dinoflagellate
- Page 9 of 61, line 152: Fig 12 is not showing 3D graph of 28, please correct to Fig 10.
- Page 9 of 61, line 158: see Fig. 10 should be read: see Fig 11b
- Page 9 of 61, line 162: is shown Table 5 should be read: is shown Table 4.
- I should suggest removing the word “strong” in the 3D graphs scale legends such as in Figure 10 and many others.
- Page 15 of 61, line 212: ciguatoxins are mentioned there when they appeared before in the manuscript. Maybe all of the ciguatoxins could be grouped together. Also, include reference at the end of such paragraph (line 217)
- Page 17 of 61, Figure legend: Karenia brevisulcata should be italic. Also revise all Figure legeds in the manuscript since italics should be applied in some cases (figure 17, Figure 20…)
- Page 18 of 61, line 230, Also include in Figure legend that gymnocin A and Gymnocin B are produced by…
- Page 24 of 61, line 270: “amphidinols (62-67)” should be read “amphidinols (66-67, Fig. 24)”
- Page 26 of 61, Figure 23. Legend says antineoplastic and antibacterial activity is shown in the graph, please correct scale legend accordingly.
- Page 26 of 61, line 281: Figure 24 is not showing structure 68-72, please correct accordingly.
- Page 26 of 61, line 282, “originally” should not be itallic.
- Page 27 of 61, Figure 24, please verify legend in Figure 24 regarding karlotoxins.
- Page 30 of 61, line 311: a suggestion to cosider writing: “Palytoxin 75 (Fig. 28 )…” instead of “Palytoxin 75 (structures see in Fig. 28)”
- Page 33 of 61. I really would like to know the criteria used by the author for choosing a particular biological activity for the 3D graphs, since the reported activities are others for some of the compouds.
- Page 34 of 61., line 376, please include reference at the end of the paragraph
- Page 39 of 61, line 460. Polyester should be corrected to polyether
- Page 42 of 61, line 480, indicate Fig 38 shows 3D graph for 108 compound. I found Figures are not always referenced in the main text and this should be revised
- Page 42 of 61, line 482 it is speaking about nigericin. Could this sentece be moved earlier in the manuscript when the author was speaking before about such compound nigericin?
- Page 42 of 61, line 501, Lasalocid 114 is the same as previously called lasalocid A (R = Me) compound number 86 in the manuscript. Therefore such sentece could be mentioned earlied in the manuscript when talking about lasalocid A and the author could consider removing compound number 114 since it is repeated.
- Page 45 of 61, line 530: “activity shown” should be read “activity prediction shown”
- Page 46 of 61, line 550: in vitro italics
- Page 50 of 61, Figure 43, please indicate photos source.
Author Response
Reviewer 2
Thanks to Reviewer 2 for reading the manuscript and some suggestions he made.
Corrected the sentences in the "Conclusion" section (sentences 619-626). To explain the PASS algorithms, a small paragraph has been inserted where all the information about PASS is given (highlighted in blue). Tables 1 and 2 compare the references where the activity information was taken. For the rest of the Tables, this is not necessary, since information on activities is presented in the text, and for many polyether ionophore antibiotics, activity was not determined.
Some minor corrections:
In the abstract, the word “polyester” is used several times. Please correct and change to polyether.
Corrected.
Page 1 of 61, line 36: “can be synthesized from… “ should be read “can be biosynthesized from… “
Corrected.
Page 3 of 61, line 67: please include references when stating “ according to published experimental data”. References are not given in the paragraph and neither in Table 1.
Corrected.
Indicate what “PASS” stands for the first time it appears in the manuscript.
Corrected.
Page 3 of 61, line 74: okadaic derivatives are shown in Table 2 (not Table 1), please correct.
Corrected.
Page 3 of 61, Figure 2. IT is stated that all photos are taken from sites where permisison in granted for non-commercial use. I belive that the author should indicate the site from where such photo was taken.
Corrected.
What is the meaning of the word “sin” in the scales in the 3D graphs??
Sin (or Cos) is a trigonometric function.
I would like to read what is the criteria that the author has followed to choose a concrete activity for the representation of the 3D graphs.
Corrected.
Include references in Table 1 and Table 2 for the reported activities. Corrected.
Page 6 of 61, the word “trans” should be italic. Corrected.
Page 6 of 61, line 127: please include reference at the end of the paragraph Corrected.
Page 7 of 61, line 132: please indicate Figure number where the ready could see structure of 19. Also consider, including compounds 18 and 19 in the same Figure since they are analogues. Corrected.
Page 8 of 61, line 139: is the legend of Figure 6 correct? Prorocenting and gambierol are also pectenotoxins? Corrected.
Page 9 of 61, line 150: dinoagellate should be read dinoflagellate Corrected.
Page 9 of 61, line 152: Fig 12 is not showing 3D graph of 28, please correct to Fig 10. Corrected.
Page 9 of 61, line 158: see Fig. 10 should be read: see Fig 11b Corrected.
Page 9 of 61, line 162: is shown Table 5 should be read: is shown Table 4. Corrected.
I should suggest removing the word “strong” in the 3D graphs scale legends such as in Figure 10 and many others. Corrected.
Page 15 of 61, line 212: ciguatoxins are mentioned there when they appeared before in the manuscript. Maybe all of the ciguatoxins could be grouped together. Also, include reference at the end of such paragraph (line 217) Corrected.
Page 17 of 61, Figure legend: Karenia brevisulcata should be italic. Also revise all Figure legeds in the manuscript since italics should be applied in some cases (figure 17, Figure 20…) Corrected.
Page 18 of 61, line 230, Also include in Figure legend that gymnocin A and Gymnocin B are produced by… Corrected.
Page 24 of 61, line 270: “amphidinols (62-67)” should be read “amphidinols (66-67, Fig. 24)” Corrected.
Page 26 of 61, Figure 23. Legend says antineoplastic and antibacterial activity is shown in the graph, please correct scale legend accordingly. Corrected.
Page 26 of 61, line 281: Figure 24 is not showing structure 68-72, please correct accordingly. Corrected.
Page 26 of 61, line 282, “originally” should not be itallic. Corrected.
Page 27 of 61, Figure 24, please verify legend in Figure 24 regarding karlotoxins. Corrected.
Page 30 of 61, line 311: a suggestion to cosider writing: “Palytoxin 75 (Fig. 28 )…” instead of “Palytoxin 75 (structures see in Fig. 28)” Corrected.
Page 33 of 61. I really would like to know the criteria used by the author for choosing a particular biological activity for the 3D graphs, since the reported activities are others for some of the compouds. Corrected.
Page 34 of 61., line 376, please include reference at the end of the paragraph Corrected.
Page 39 of 61, line 460. Polyester should be corrected to polyether Corrected.
Page 42 of 61, line 480, indicate Fig 38 shows 3D graph for 108 compound. I found Figures are not always referenced in the main text and this should be revised Corrected.
Page 42 of 61, line 482 it is speaking about nigericin. Could this sentece be moved earlier in the manuscript when the author was speaking before about such compound nigericin? Corrected.
Page 42 of 61, line 501, Lasalocid 114 is the same as previously called lasalocid A (R = Me) compound number 86 in the manuscript. Therefore such sentece could be mentioned earlied in the manuscript when talking about lasalocid A and the author could consider removing compound number 114 since it is repeated. Corrected.
Page 45 of 61, line 530: “activity shown” should be read “activity prediction shown” Corrected.
Page 46 of 61, line 550: in vitro italics Corrected.
Page 50 of 61, Figure 43, please indicate photos source. Corrected.
All changes and additions are made right in the text and are highlighted in yellow.
Valery Dembitsky
Reviewer 3 Report
The manuscript is extremely long, mainly because there are too many figures and tables. This makes the entire reading process a bit tiresome. It will become more interesting if these are reduced. I would suggest keeping a representative structure from each metabolite type, and their derivatives could be moved as supplementary figures.
Similarly, for 3D graphs, a single figure could be provided by combining multiple figures from the same group of compounds.
The tables too have redundant information. All tables in the current manuscript are row-based. That means for each compound, the same activity is mentioned repeatedly. For example, in Table 2, DNA synthesis inhibitor is repeated for compounds 2, 3, 4, and 5. Authors could reformat the tables to a column-based format where the first row/column could be the list of redundant activities. Moreover, in my opinion, instead of so many tables in the main manuscript, if possible it will be much better and more informative to have a single table (even if it's big) that summarizes the entire review.
I could not find any reference or link to the PASS software in the review. Moreover, some overview about the methodology adopted by the author should be briefly provided, e.g. What input is accepted by PASS? How was the input prepared? Etc.
Species name should be in italics, e.g. Figures 16, 17, etc.
Page 26, Line 280-289: Font size for organism names is different than the text.
Page 39 Line 457: So eel2????
Page 44 Line 524: Streptomyces is not a fungus.
Author Response
Reviewer 3
Thanks to Reviewer 3 for reading the manuscript and some suggestions he made.
Yes, I agree with the reviewer that the article is long and contains many tables and figures, and it is apparently tiring to read. However, this is an overview. And as you know, reviews are never small. Regarding the merging of charts and structures, this is not realistic as this overview has been halved compared to the original version. The tables contain only the information that correlates with her activity. Other adjustments or changes are unnecessary.
Regarding the PASS program that the reviewer is interested in. A small paragraph has been inserted where all the information about PASS is presented.
Species name should be in italics, e.g. Figures 16, 17, etc. Corrected.
Page 26, Line 280-289: Font size for organism names is different than the text. Corrected.
Page 39 Line 457: So eel2???? Corrected.
Page 44 Line 524: Streptomyces is not a fungus. Corrected.
All changes and additions are made right in the text and are highlighted in blue.
Valery Dembitsky
Round 2
Reviewer 1 Report
Although the author has revised the manuscript, the primary issue remains unresolved. Regardless of the type of manuscript to which it belongs, my only concern is that if QSAR was used, it should be described under the methodology section, documenting all procedures undertaken.
Author Response
Reviewer 1
Thanks to Reviewer 1 for reading the manuscript and some suggestions he made.
After reading your comments, I concluded that you are not an expert in the area under discussion. Since no specific comments were submitted! Except for one thing, this is not a review article.
I repeat once again that I fundamentally disagree with reviewer 1's remark, which was confirmed by two other reviewers who agreed that the submitted article is a review. I adhere to the opinion of reviewers 2 and 3!
This is a typical review article describing the biological activity of natural polyester ionophores. In addition to the already known activities, an additional activity obtained using PASS has been added.
To accommodate reviewer 1's request, a short paragraph has been inserted with all the information about QSAR. We hope that this information will be sufficient for understanding the presented review, its figures and tables.
Yes, I agree, the title of the review is not very good, it has been changed.
All changes and additions are made right in the text and are highlighted in blue.